# Robust Multi-Agent Reinforcement Learning with Model Uncertainty

**Kaiqing Zhang**[*♮]   **Tao Sun**[†♮]   **Yunzhe Tao**[†]   **Sahika Genc**[†]   **Sunil Mallya**[†]   **Tamer Başar**[*]

[*]Department of ECE and CSL, University of Illinois at Urbana-Champaign

[†]Amazon Web Services

{kzhang66, basar1}@illinois.edu

{suntao, yunzhet, sahika, smallya}@amazon.com

## Abstract

In this work, we study the problem of multi-agent reinforcement learning (MARL) with model uncertainty, which is referred to as *robust MARL*. This is naturally motivated by some multi-agent applications where each agent may not have perfectly accurate knowledge of the model, e.g., all the reward functions of other agents. Little *a priori* work on MARL has accounted for such uncertainties, neither in problem formulation nor in algorithm design. In contrast, we model the problem as a *robust Markov game*, where the goal of all agents is to find policies such that no agent has the incentive to deviate, i.e., reach some equilibrium point, which is also robust to the possible uncertainty of the MARL model. We first introduce the solution concept of robust Nash equilibrium in our setting, and develop a Q-learning algorithm to find such equilibrium policies, with convergence guarantees under certain conditions. In order to handle possibly enormous state-action spaces in practice, we then derive the policy gradients for robust MARL, and develop an actor-critic algorithm with function approximation. Our experiments demonstrate that the proposed algorithm outperforms several baseline MARL methods that do not account for the model uncertainty, in several standard but uncertain cooperative and competitive MARL environments.

## 1 Introduction

Deep reinforcement learning (RL) has recently achieved tremendous successes in many sequential decision-making problems, varying from robotics [1, 2] and autonomous driving [3] to game playing [4, 5]. In fact, many of these important applications involve more than one agent or player [6, 7], naturally leading to the setting of multi-agent RL (MARL). MARL addresses the decision-making problem of multiple agents in a common environment, where the goal of each agent is to optimize its own long-term return by interacting with the environment and other agents; see [8, 9] for detailed reviews, and [10, 11, 12, 13, 14, 15] for some recent advances. MARL problems are usually modeled under the framework of *Markov games*, stemming from the seminal work [16].

In real-world applications, the agents, especially those trained in simulations, may not have perfectly accurate knowledge of the actual model, i.e., the reward functions of all agents and the transition probability model. In particular, the solution obtained from the simulation without uncertainty may have poor performance in practice, known as the *sim-to-real* gap. Such an issue has been reported quite common in the autonomous-car racing application [17], which initially motivates the present work. In single-agent RL, such an uncertainty has been nicely handled through the lens of *robust Markov decision processes* (MDPs) [18, 19, 20] and *robust (adversarial) RL* [21, 22]. In comparison, such an uncertainty has not been fully explored in the multi-agent RL regime. In light

---

[♮]Equal Contribution

of the significance and ubiquity of MARL, it is thus imperative to take the model uncertainty into account in both the *formulation* and the *algorithm design* in this setting.

In this work, we aim to develop such a *robust MARL* framework when model uncertainty is present. Specifically, we model the problem of MARL with model uncertainty as a *robust* Markov game [23], where the goal of all agents is to find policies such that no agent has the incentive to deviate, which are also robust to the possible uncertainty of the model. All agents play a standard Markov game with additional concerns on the distribution-free uncertainty of the reward function and transition probabilities. To adapt to the worst-case scenario due to uncertainty, one can view the uncertainty as the decision made by an implicit player, a "nature" player, who always plays against each agent. This way, the solution concept in robust Markov games differs from the standard Nash equilibrium (NE) in Markov games [24]. Indeed, each agent not only needs to optimize its own return, in consideration of other agents' general affects on itself, but also needs to always play against the nature player, in order to tackle model uncertainty. Such a model covers MARL environments with both cooperative and competitive agents. Within this new framework, we then develop several robust MARL algorithms to find the solution concept of the game, and evaluate their performance in benchmark MARL environments. We summarize our contribution as follows.

**Contribution.**   Our contribution is three-fold: first, we propose a new framework to systematically characterize the model uncertainty in MARL, by advocating the use of robust Markov games; second, we develop both Q-learning-based and actor-critic algorithms for finding the solution concept in this framework; third, we validate the performance of our algorithms with function approximation and mini-batch update, via extensive simulations in benchmark MARL environments. To the best of our knowledge, we provide the first formulation and algorithms that account for the model uncertainties in MARL, with both theoretical and empirical justifications.

**Related Work.**   Our work falls into the regime of MARL that originates from the seminal work [16], under the framework of Markov games [24]. Going beyond the zero-sum setting in [16], [25, 26, 27] have considered general-sum Markov games. Most of the later MARL works, either empirical or theoretical, have been built upon this Markov game model, e.g., [14, 13, 28, 29, 30, 31]. Despite the numerous advances in MARL recently, however, few of them based on Markov games have handled the uncertainty in the model, which is the focus of our work. The closest setting to ours is the recent work [32], which also considered robustness in MARL problems. Nonetheless, we highlight that the robustness there is with respect to the changes of the *opponents' policies*, between the training and testing phases, instead of the robustness to the model uncertainty that we consider here.

Model uncertainty has been nicely handled in single-agent RL. Notably, one classical and rigorous formulation of robust RL is the robust MDP framework [18, 19, 20], where the model uncertainty is treated as an adversary that plays against the agent, leading to a two-agent zero-sum game. Robust RL algorithms were then developed for this setting in [21, 33, 34, 35]. Such a zero-sum game/minimax formulation has also been adopted in other works [36, 22, 37, 38], in order to handle the sim-to-real gap. Besides this worst-case modeling, [39] also considered a distributional framework to model uncertainty in MDPs, and [40] recently proposed distributionally robust RL algorithms. However, it is not yet clear how these approaches can be generalized to multi-agent settings. In fact, with an additional adversary in MARL, the underlying model is no longer two-agent zero-sum, but falls into the *general-sum* regime, which is much harder to solve in general [41], or develop RL algorithms for [25, 26, 27]. Motivated by the robust Markov game model in operations research [23], we attempt to make an initial step toward this direction for robust MARL.

## 2   Problem Formulation

In this section, we provide the formulation of robust MARL problems, and the background on some fundamental concepts in the setting.

### 2.1   Markov Games and MARL

We model the interaction among agents in a general framework, i.e., *Markov games* [16]. A Markov game $\mathcal{G}$ is a tuple

$$\mathcal{G} := \langle \mathcal{N}, \mathcal{S}, \{\mathcal{A}^i\}_{i \in \mathcal{N}}, \{R^i\}_{i \in \mathcal{N}}, P, \gamma \rangle,$$

where $\mathcal{N} = [N]$ denotes the set of $N$ agents, $\mathcal{S}$ is the state space that is shared by all agents, and $\mathcal{A}^i$ denotes the action space of agent $i \in \mathcal{N}$. $R^i : \mathcal{S} \times \mathcal{A}^1 \times \cdots \times \mathcal{A}^N \to \mathbb{R}$ represents the reward function of agent $i$, which depends on the state and the joint action of all agents. $P : \mathcal{S} \times \mathcal{A}^1 \times \cdots \times \mathcal{A}^N \to \Delta(\mathcal{S})$ represents the state transition probability that is a mapping from the current state and the joint action to the probability distribution over the state space. Lastly, $\gamma \in [0, 1)$ is the discounting factor.

At time $t$, each agent selects its own action $a_t^i \in \mathcal{A}^i$ given the system state $s_t$, according to its own policy $\pi^i : \mathcal{S} \to \Delta(\mathcal{A}^i)$, which is a mapping from the state space to the probability distribution over action space $\mathcal{A}^i$. Note that here we only consider the *Markov policies* that depend on the current state $s_t$ at time $t$. Then, the system transits to the next state $s_{t+1}$ and each agent $i$ receives an instantaneous reward $r_t^i = R^i(s_t, a_t^1, \cdots, a_t^N)$. The goal of each agent $i$ is to maximize the long-term return $J^i$ calculated by:

$$\max_{\pi^i} \ J^i(\pi^i, \pi^{-i}) := \mathbb{E}\bigg[ \sum_{t=0}^{\infty} \gamma^t r_t^i \ \bigg| \ s_0, a_t^i \sim \pi^i(\cdot \, | \, s_t), a_t^{-i} \sim \pi^{-i}(\cdot \, | \, s_t) \bigg],$$

where $-i$ represents the indices of all agents except agent $i$, and $\pi^{-i} := \prod_{j \neq i} \pi^j$ refers to the joint policy of all agents except agent $i$. In the same vein, one can define the value and action-value (Q-)function for each agent $i$ as

$$V^i(s) := \mathbb{E}\bigg[ \sum_{t=0}^{\infty} \gamma^t r_t^i \ \bigg| \ s_0 = s, a_t^i \sim \pi^i(\cdot \, | \, s_t), a_t^{-i} \sim \pi^{-i}(\cdot \, | \, s_t) \bigg]$$

and

$$Q^i(s, a^1, \cdots, a^N) := \mathbb{E}\bigg[ \sum_{t=0}^{\infty} \gamma^t r_t^i \ \bigg| \ s_0 = s, a_0^i = a^i, a_0^{-i} = a^{-i}, a_t^i \sim \pi^i(\cdot \, | \, s_t), a_t^{-i} \sim \pi^{-i}(\cdot \, | \, s_t) \bigg].$$

Since agents' policies are coupled in $J^i$, maximizing the return of a single agent is unattainable without considering the policies of other agents. Instead, one commonly used solution concept is the (Markov perfect) Nash equilibrium of the game. The NE is defined as the point of a joint policy $\pi_* := (\pi_*^1, \cdots, \pi_*^N)$ at which for any policy $\pi^i$

$$J^i(\pi_*^i, \pi_*^{-i}) \geq J^i(\pi^i, \pi_*^{-i}), \quad \forall i \in \mathcal{N},$$

namely, given all other agents' equilibrium policies $\pi_*^{-i}$, there is no motivation for agent $i$ to deviate from $\pi_*^i$. The goal of most MARL problems is to solve for the NE of the Markov games $\mathcal{G}$ without the knowledge of the model.

## 2.2 Robust Markov Games

In many practical applications, the agents may not have perfect knowledge of the model, i.e., the reward function and/or the transition probability model. For example, in an urban traffic network that involves multiple self-driving cars, each vehicle makes an individual action and cannot have perfect knowledge of other cars' rewards and the exact joint transition model. Thus, the desired policy should not only be able to play against other agents' policies, but also robust to the possible uncertainty of the MARL model. Formally, this problem can be modeled as a *robust Markov game*, or equivalently, robust stochastic game [23], which is characterized by the following tuple:

$$\bar{\mathcal{G}} := \langle \mathcal{N}, \mathcal{S}, \{\mathcal{A}^i\}_{i \in \mathcal{N}}, \{\bar{\mathcal{R}}_s^i\}_{(i,s) \in \mathcal{N} \times \mathcal{S}}, \{\bar{\mathcal{P}}_s\}_{s \in \mathcal{S}}, \gamma \rangle,$$

where $\mathcal{N}$, $\mathcal{S}$, $\mathcal{A}^i$, and $\gamma \in [0, 1)$ denote the set of agents, the state space, the action space for each agent $i$, and the discounting factor, respectively. For notation convenience, let $\mathcal{A} := \mathcal{A}^1 \times \cdots \times \mathcal{A}^N$. Then we denote by $\bar{\mathcal{R}}_s^i \subseteq \mathbb{R}^{|\mathcal{A}|}$ and $\bar{\mathcal{P}}_s$ the uncertainty sets of all possible reward function values and that of all possible transition probabilities at state $s$, respectively. Note that the uncertainty set for the reward function $\bar{\mathcal{R}}_s^i$ may vary for different agent $i$.

Each player considers a distribution-free Markov game to be played using robust optimization. The formulation allows the use of simple uncertainty sets of the model, and requires no *a prior* probabilistic information about the uncertainty, e.g., distribution of the class of models. Note that if the player knows how to play in the robust Markov game optimally starting from the next stage on, then it would play to maximize not only the worst-case (minimal) expected immediate reward, due to the model uncertainty set at the current stage, but also the worst-case expected reward incurred in the

future stages. Formally, such a recursion property leads to the following Bellman-type equation:

$$\bar{V}_*^i(s) = \max_{\pi^i(\cdot\,|\,s)} \min_{\substack{\bar{R}_s^i \in \bar{\mathcal{R}}_s^i \\ \bar{P}(\cdot\,|\,s,\cdot) \in \bar{\mathcal{P}}_s}} \sum_{a \in \mathcal{A}} \prod_{j=1}^N \pi^j(a^j\,|\,s)\left(\bar{R}^i(s,a) + \gamma \sum_{s' \in \mathcal{S}} \bar{P}(s'\,|\,s,a)\bar{V}_*^i(s')\right), \quad (2.1)$$

where $\bar{V}_*^i : \mathcal{S} \to \mathbb{R}$ denotes the *optimal robust value*, and $\bar{R}_s^i = [\bar{R}^i(s,a)]_{a \in \mathcal{A}}^\top \in \bar{\mathcal{R}}_s^i \subseteq \mathbb{R}^{|\mathcal{A}|}$ with $a = (a^1, \cdots, a^N)$, is the vector of possible reward values of agent $i$ that lies in the uncertain set of vectors $\bar{\mathcal{R}}_s^i$ at state $s$. $\bar{P}(\cdot\,|\,s,\cdot) : \mathcal{A} \to \Delta(\mathcal{S})$ denotes the possible transition probability lying in the uncertain set $\bar{\mathcal{P}}_s$.

The uncertainty here can be viewed as the decision made by an implicit player, *the nature*, who always plays against each agent $i$ by selecting the worst-case model data at every state. If such an optimal robust value exists, then it leads to the definition of robust Markov perfect Nash equilibrium (RMPNE), the solution concept for robust Markov games, as follows.

**Definition 2.1.** A joint policy $\pi_* = (\pi_*^1, \pi_*^2, \cdots, \pi_*^N)$ is the *robust Markov perfect Nash equilibrium*, if for any $s \in \mathcal{S}$ and all $i \in \mathcal{N}$, there exists a vector of value functions $\bar{V}_* = (\bar{V}_*^1, \cdots, \bar{V}_*^N)$ with each $\bar{V}_*^i : \mathcal{S} \to \mathbb{R}$ satisfying

$$\pi_*^i(\cdot\,|\,s) \in \underset{\pi^i(\cdot\,|\,s)}{\mathrm{argmax}} \min_{\substack{\bar{R}_s^i \in \bar{\mathcal{R}}_s^i \\ \bar{P}(\cdot\,|\,s,\cdot) \in \bar{\mathcal{P}}_s}} \sum_{a \in \mathcal{A}} \pi^i(a^i\,|\,s) \prod_{j \neq i} \pi_*^j(a^j\,|\,s)\left(\bar{R}^i(s,a) + \gamma \sum_{s' \in \mathcal{S}} \bar{P}(s'\,|\,s,a)\bar{V}_*^i(s')\right).$$

By Definition 2.1, the RMPNE we consider is *stationary*, i.e., time-invariant. We now verify that such an NE exists.

**Proposition 2.2.** Suppose that the state and action spaces $\mathcal{S}$ and $\mathcal{A}$ are finite, and the uncertain sets of both the transition probabilities and rewards of the robust Markov game $\bar{\mathcal{G}}$, namely, $\bar{\mathcal{P}}_s$ and $\bar{\mathcal{R}}_s^i$ for all $s \in \mathcal{S}$ and $i \in \mathcal{N}$, belong to compact sets. Then, a robust Markov perfect Nash equilibrium exists.

The proof of Proposition 2.2 is deferred to §A.1. Without loss of generality, we follow the convention as in [23], and only focus on the uncertainty in the reward function hereafter for simplicity. Namely, the set $\bar{\mathcal{P}}_s = \{P(\cdot\,|\,s,\cdot)\}$ only has one element, the exact transition $P(\cdot\,|\,s,\cdot)$. The approach can be extended to consider the uncertainty in both rewards and transition dynamics, as argued in [23]. Define $T^i(s,a) = \bar{R}^i(s,a) + \gamma \sum_{s' \in \mathcal{S}} P(s'\,|\,s,a)\bar{V}_*^i(s')$. Then the Bellman-type equation in (2.1) can be written as

$$\bar{V}_*^i(s) = \max_{\pi^i(\cdot\,|\,s)} \min_{\bar{R}_s^i \in \bar{\mathcal{R}}_s^i} \sum_{a \in \mathcal{A}} \pi^i(a^i\,|\,s) \prod_{j \neq i} \pi_*^j(a^j\,|\,s) T^i(s,a) \quad (2.2)$$

$$= \max_{\pi^i(\cdot\,|\,s)} \min_{\pi^{0,i}(\cdot\,|\,s)} \mathbb{E}_{\bar{R}_s^i \sim \pi^{0,i}(\cdot\,|\,s), a^i \sim \pi^i(\cdot\,|\,s), a^{-i} \sim \pi_*^{-i}(\cdot\,|\,s)} T^i(s,a), \quad (2.3)$$

where $\pi^{0,i}(\cdot\,|\,s) \in \Delta(\bar{\mathcal{R}}_s^i)$ denotes the policy of the nature against agent $i$ at state $s$, a probability distribution over the uncertain set $\bar{\mathcal{R}}_s^i$ of agent $i$'s reward. The nature's policy against each agent $i$ can be different because the agents are not necessarily symmetric, namely, the reward uncertainty and the role they play in the transition $P$ may be different. Thus, the policy for the nature is in fact a set, i.e., $\pi^0 := \{\pi^{0,i}\}_{i \in \mathcal{N}}$. The equivalence between (2.2) and (2.3) is due to the fact that the inner-loop minimization over a deterministic choice of $\bar{R}_s^i$ can be achieved by minimizing over the stochastic strategy, i.e., a probability distribution, over the support $\bar{\mathcal{R}}_s^i$.

Using the nature player and its policy set $\pi^0 = \{\pi^{0,i}\}_{i \in \mathcal{N}}$, we further define the solution concept of *RMPNE with nature* (NRMPNE) as follows.

**Definition 2.3.** A joint policy $\tilde{\pi}_* = (\pi^0_*, \pi^1_*, \cdots, \pi^N_*)$ is the *robust Markov perfect Nash equilibrium with nature*, where $\pi^0_* = \{\pi_*^{0,i}\}_{i \in \mathcal{N}}$, if for any $s \in \mathcal{S}$ and all $i \in \mathcal{N}$, there exists a vector of value functions $\bar{V}_* = (\bar{V}_*^1, \cdots, \bar{V}_*^N)$ with each $\bar{V}_*^i : \mathcal{S} \to \mathbb{R}$, such that

$$\left(\pi_*^i(\cdot\,|\,s), \pi_*^{0,i}(\cdot\,|\,s)\right) \in \underset{\pi^i(\cdot\,|\,s)}{\mathrm{argmax}} \min_{\pi^{0,i}(\cdot\,|\,s)} \sum_{\bar{R}_s^i \in \bar{\mathcal{R}}_s^i} \pi^{0,i}\left(\bar{R}_s^i\,|\,s\right) \sum_{a \in \mathcal{A}} \pi^i(a^i\,|\,s) \prod_{j \neq i} \pi_*^j(a^j\,|\,s) T^i(s,a),$$

where we recall that $\bar{R}_s^i = [\bar{R}^i(s,a)]_{a \in \mathcal{A}}^\top$.

By Proposition 2.2, the existence of an RMPNE $\pi_*$ is equivalent to the existence of an NRMPNE $\widetilde{\pi}_*$ with $\pi_*^{0,i}$ being a *deterministic* policy for all $i$. Therefore, hereafter we will only consider the nature's policy as a deterministic one, namely, for each $s \in \mathcal{S}$, $\pi_*^{0,i}(s) = \bar{R}_s^i \in \bar{\mathcal{R}}_s^i$.

## 3 Algorithms

To find the robust Markov perfect NE defined in Definition 2.1, or equivalently in Definition 2.3, one has to solve the Bellman-type fixed-point equation for the robust Markov game in (2.2), or (2.3). In this section, we first develop a value iteration approach, when the model is known to the agents. Based on this, we then propose a model-free Q-learning-based algorithm with convergence guarantees under certain conditions. In addition, viewing the nature as another agent, we also develop a policy gradient-based method with function approximation.

### 3.1 Value Iteration and Q-learning for Robust MARL

By Bellman equation (2.2), one straightforward approach is to develop *value iteration* (VI) algorithms when the model on $\bar{\mathcal{G}}$ is known. In particular, the goal is to learn a value function $\bar{V}$ by updating the Bellman equation (2.2) recursively such that for all $i \in \mathcal{N}$:

$$\bar{V}_{t+1}^i(s) = \max_{\pi^i(\cdot \,|\, s)} \; \min_{\bar{R}_s^i \in \bar{\mathcal{R}}_s^i} \; \sum_{a \in \mathcal{A}} \prod_{j=1}^N \pi^j(a^j \,|\, s) T^i(s, a) =: \mathcal{T}_{\mathcal{V}}^i(\bar{V}_t^i). \tag{3.1}$$

As a result, the desired value function $\bar{V}^i$ is a fixed-point of the operator $\mathcal{T}_{\mathcal{V}}^i(\cdot) : \mathbb{R}^{|\mathcal{S}|} \to \mathbb{R}^{|\mathcal{S}|}$.

Building upon the VI update in (3.1), one can develop Q-learning-based algorithms. In particular, the equilibrium action-value function, i.e., Q-value function, of robust Markov games can be written as a function of state, joint action, and reward, which satisfies the following Bellman equation:

$$\bar{Q}_*^i\big(s, a, \bar{R}^i(s, a)\big) := \bar{R}^i(s, a) + \gamma \sum_{s' \in \mathcal{S}} P(s' \,|\, s, a) \sum_{a'} \bigg( \prod_{j=1}^N \pi_*^j(a'^j \,|\, s') \bigg) \bar{Q}_*^i\big(s', a', \bar{R}_*^i(s', a')\big), \tag{3.2}$$

where $a = (a^1, \cdots, a^N)$ and $a' = (a'^1, \cdots, a'^N)$, $\pi_*^j$ is the equilibrium policy of agent $j$, and $\bar{R}_*^i(s', a')$ is the $a'$-th element of $\bar{R}_{*,s'}^i = \pi_*^{0,i}(s')$, the output of the nature's deterministic policy at the equilibrium. Note that different from the standard Bellman equation for single-agent Q-value function, the equilibrium policy $\pi_*$ cannot be obtained just from $\bar{Q}^i$ (which is the greedy policy for the single-agent setting). In fact, the Q-values of all other agents are also required to determine the equilibrium policy $\pi_*$, which is the challenging part in developing multi-agent Q-learning algorithms in general [25, 26], compared to the single-agent Q-learning.

As a consequence, the tabular-setting Q-learning update can be written as

$$\bar{Q}_{t+1}^i(s_t, a_t, \bar{R}_t^i) := (1 - \alpha_t) \cdot \bar{Q}_t^i(s_t, a_t, \bar{R}_t^i) + \alpha_t \cdot \bigg[ \bar{R}_t^i + \gamma \sum_{a_{t+1}} \pi_{*,t}(a_{t+1} \,|\, s_{t+1}) \bar{Q}_t^i(s_{t+1}, a_{t+1}, \bar{R}_{t+1}^i) \bigg], \tag{3.3}$$

with $\bar{R}_t^i = \pi_{*,t}^{0,i}(s_t)[a_t]$, $a_t^i \sim \pi_{*,t}^i(\cdot \,|\, s_t)$, and $\bar{R}_{t+1}^i = \pi_{*,t}^{0,i}(s_{t+1})[a_{t+1}]$. Here, $\pi_{*,t}^0 = \big\{\pi_{*,t}^{0,i}\big\}_{i \in \mathcal{N}}$ and $\pi_{*,t} = \prod_{j=1}^N \pi_{*,t}^j$ denote the equilibrium policies of the nature and the equilibrium joint policies of all agents, respectively, computed from $\{\bar{Q}_t^i\}_{i \in \mathcal{N}}$ at time $t$. The term $\pi_{*,t}^{0,i}(s)[a]$ denotes the $a$-th element of the policy output $\pi_{*,t}^{0,i}(s)$, a real vector that lies in $\bar{\mathcal{R}}_s^i \subseteq \mathbb{R}^{|\mathcal{A}|}$.

**Convergence.** Note that convergence of the update (3.3) is in general hard to establish, as the Bellman operator induced by solving a general-sum game in (3.2) does not always satisfy the conditions for the convergence of Q-learning in MDPs and generalized MDPs [42]. As recognized in [25, 26, 27], convergence of Q-learning in general-sum Markov games indeed requires more conditions. We will establish the convergence of (3.3) under certain conditions, mostly motivated from [25]. Due to space limitation, we defer the results in Supplementary §A.2. The results, though not generally apply to all robust Markov games, provide some proof-of-concept justifications and sanity-check for the convergence of the value-based/Q-learning update. Indeed, developing provable convergent

Q-learning for general-sum Markov games without restrictive assumptions remains open, and is still worth further investigation.

Note that (3.3) needs to maintain the Q-value iterate of all agents, which increases the complexity of the algorithm, but is inevitable for value-based RL algorithms for general-sum Markov games. This is mainly due to the fact that at each iteration, the Q-value estimates of all agents are required in order to perform one-step of the sample-based equilibrium computation. The same issue also occurs in the design of the Nash-Q learning algorithm [25]. Another issue is that it is computationally hard to calculate the equilibrium at each iteration, even if the payoff matrix $\left( \bar{Q}_t^1(\cdot \,|\, s_{t+1}), \cdots, \bar{Q}_t^N(\cdot \,|\, s_{t+1}) \right)$ is given. The complexity of solving for this equilibrium in general-sum games can be high [41]. Finally, it is not clear yet how the Q-learning update can be incorporated with function approximation scheme, to handle extremely large state-action spaces in practice. As a consequence, we are motivated to develop policy-gradient/actor-critic-based robust MARL algorithms, as to be introduced next.

## 3.2   Policy Gradient/Actor-Critic for Robust MARL

In contrast to the value-based methods, policy-gradient/actor-critic methods can easily incorporate function approximation into the update, in order to handle massive or even continuous state-action spaces. Such an upshot is even more significant and necessary in multi-agent RL, as the joint action space grows exponentially with the number of agents.

In particular, each agent $i$'s policy $\pi^i$ is parameterized as[2] $\pi_{\theta^i}$ for $i \in \mathcal{N}$, and the nature's policy is parameterized by a set of policies $\pi_{\theta^0} := \{\pi_{\theta^{0,i}}\}_{i \in \mathcal{N}}$. Note that we here parameterize all $\pi_{\theta^{0,i}}$ as *deterministic* policies, i.e., $\pi_{\theta^{0,i}}(s) = \bar{R}_s^i \in \mathcal{R}_s^i$, due to Proposition 2.2. Also, we define the return objective of each agent $i$ under the joint policy $\widetilde{\pi}_\theta := (\pi_{\theta^0}, \pi_{\theta^1}, \cdots, \pi_{\theta^N})$ as $J^i(\theta) := \bar{V}_{\widetilde{\pi}_\theta}^i(s_0)$, where $s_0$ denotes the initial state[3], $\theta = (\theta^0, \theta^1, \cdots, \theta^N)$ is the concatenation of all policy parameters with $\theta^0 := (\theta^{0,1}, \cdots, \theta^{0,N})$, and $\bar{V}_{\widetilde{\pi}_\theta}^i$ is the value function under joint policy $\widetilde{\pi}_\theta$ that satisfies

$$\bar{V}_{\widetilde{\pi}_\theta}^i(s) = \sum_{a \in \mathcal{A}} \prod_{j=1}^N \pi_{\theta^j}(a^j \,|\, s) \left( \pi_{\theta^{0,i}}(s)[a] + \gamma \sum_{s' \in \mathcal{S}} P(s' \,|\, s, a) \bar{V}_{\widetilde{\pi}_\theta}^i(s') \right), \qquad (3.4)$$

where $\pi_{\theta^{0,i}}(s)[a]$ is the $a$-th element of the output vector $\pi_{\theta^{0,i}}(s)$. Similarly, we also define the Q-value under joint policy $\widetilde{\pi}_\theta$ to be the one that satisfies the fixed-point equation

$$\bar{Q}_{\widetilde{\pi}_\theta}^i(s, a) = \pi_{\theta^0}(s)[a] + \gamma \sum_{s' \in \mathcal{S}} P(s' \,|\, s, a) \sum_{a' \in \mathcal{A}} \prod_{j=1}^N \pi_{\theta^j}(a'^j \,|\, s') \bar{Q}_{\widetilde{\pi}_\theta}^i(s', a'). \qquad (3.5)$$

We first establish the general policy gradient with respect to the parameter $\theta$ in the following lemma.

**Lemma 3.1** (Policy Gradient Theorem in Robust MARL). *For each agent* $i = 1, \cdots, N$, *the policy gradient of the objective* $J^i(\theta)$ *with respect to the parameter* $\theta$ *has the following form[4]*:

$$\nabla_{\theta^i} J^i(\theta) = \mathbb{E}_{s \sim \rho_{\pi_\theta}^{s_0}, a \sim \pi_\theta(\cdot \,|\, s)} \left[ \nabla \log \pi_{\theta^i}(a^i \,|\, s) \bar{Q}_{\widetilde{\pi}_\theta}^i(s, a) \right], \qquad (3.6)$$

$$\nabla_{\theta^{0,i}} J^i(\theta) = \mathbb{E}_{s \sim \rho_{\pi_\theta}^{s_0}, a \sim \pi_\theta(\cdot \,|\, s)} \left[ \nabla \pi_{\theta^{0,i}}(s)[a] \right], \qquad (3.7)$$

*where* $\pi_\theta(a \,|\, s) := \prod_{j=1}^N \pi_{\theta^j}(a^j \,|\, s)$, $\rho_{\pi_\theta}^{s_0}(s) := \sum_{t=0}^\infty \gamma^t \cdot Pr(s_0 \to s, t, \pi_\theta)$ *is the discounted state visitation measure under joint policy* $\pi_\theta$ *with state starting from* $s_0$, *with* $Pr(s \to s', t, \pi_\theta)$ *denoting the probability of transitioning from* $s$ *to* $s'$ *under joint policy* $\pi_\theta$ *with* $t$-*steps, and* $\pi_{\theta^{0,i}}(s)[a]$ *is the* $a$-*th element of the output of* $\pi_{\theta^{0,i}}(s)$.

The proof can be found in Supplementary §B. Note that the visitation measure $\rho_{\pi_\theta}^{s_0}$ only depends on the joint policy of all agents, i.e., $\pi_\theta$, but not the nature's policy $\pi_{\theta^0}$. Moreover, we note that the form

in (B.2) bears some resemblance with the standard policy gradient theorem [43], with the Q-function being replaced by the robust one under the joint policy $\widetilde{\pi}_\theta$. In contrast, the form in (B.3) does not involve the Q-function, and is more similar to the deterministic policy gradient theorem [44]. Finally, we note that the statement in §B is more general, where the transition model $P$ is also parametrized and viewed as uncertain. Similarly, one can view the parametrized transition as the policy of the nature, and always play against all agents. Thus in Lemma B.1 in §B, we also include the policy gradient with respect to the transition model parameter for completeness.

In particular, if the robust Q-value function $\bar{Q}^i_{\pi_\theta}$ is also parameterized as $\bar{Q}_{\omega^i} : \mathcal{S} \times \mathcal{A} \to \mathbb{R}$ by some parameter $\omega^i \in \mathbb{R}^d$. Then, some critic algorithm, e.g., the temporal difference (TD) learning algorithm, can be applied to evaluate the joint policy $\widetilde{\pi}_\theta$. This naturally gives us the online actor-critic algorithm as follows:

**Critic:** $\delta^i_t = \pi_{\theta^{0,i}_t}(s_t)[a_t] + \gamma \bar{Q}_{\omega^i_t}(s_{t+1}, a_{t+1}) - \bar{Q}_{\omega^i_t}(s_t, a_t), \quad \omega^i_{t+1} = \omega^i_t + \alpha_t \cdot \delta^i_t \cdot \nabla \bar{Q}_{\omega^i_t}(s_t, a_t),$

**Actor:** $\theta^i_{t+1} = \theta^i_t + \beta_t \cdot \nabla \log \pi_{\theta^i_t}(a^i_t \mid s_t) \cdot \bar{Q}_{\omega^i_t}(s_t, a_t), \qquad \theta^{0,i}_{t+1} = \theta^{0,i}_t - \beta_t \cdot \nabla \pi_{\theta^{0,i}_t}(s_t)[a_t],$

where $\delta^i_t$ is the TD error for agent $i$, $\alpha_t, \beta_t > 0$ are both step sizes that may diminish over time, namely, $\lim_{t\to\infty} \alpha_t = \lim_{t\to\infty} \beta_t = 0$, and also satisfy $\sum_{t\geq 0} \alpha_t^2 < \infty, \sum_{t\geq 0} \beta_t^2 < \infty, \sum_{t\geq 0} \alpha_t = \sum_{t\geq 0} \beta_t = \infty$. Moreover, $\alpha_t$ is usually larger than $\beta_t$ as $t \to \infty$, i.e., $\lim_{t\to\infty} \beta_t/\alpha_t = 0$, in order to ensure that the critic step performs faster than the actor step. This is also known as a *two-timescale* actor-critic algorithm [45, 46]. In practice, both critic and actor can be updated in a mini-batch fashion [47, 13, 48]. See Algorithm 1 in Supplementary §C for the pseudo-code of our actor-critic-based robust MARL algorithm. In designing the algorithm, we adopt a centralized-training-decentralized-execution paradigm, following the popular MARL framework in [13].

## 4 Experimental Results

To demonstrate the effectiveness of the proposed algorithm, we provide experimental results in several benchmark competitive and cooperative MARL environments, based on the multi-agent particle environments developed in [13]. Specifically, we consider the cooperative navigation, keep-way, physical deception, and predator-prey environments. Detailed description of the experimental setting can be found in Supplementary §D. We directly compare the performance of our algorithm with MADDPG [13], where no robustness is considered, and M3DDPG [32], where robustness is considered with respect to the *changes of the opponents' policies*, instead of the model uncertainty. Although M3DDPG was not designed to handle this uncertainty, we compare with it in the cooperative navigation and keep-away experiments for completeness, as some reviewers have suggested.

In order to test the robustness of the proposed algorithm, which is referred to as *Robust-MADDPG*, or *R-MADDPG* for brevity, we impose different levels of uncertainty to the rewards returned from each particle environment. In particular, we use truncated Gaussian noise, defined as $\bar{R}(s, a) = \mathcal{N}_{\text{trunc}}(R(s, a), \lambda)$, to ensure the compactness of the uncertainty set. The parameter $\lambda$ controls the uncertainty level of the rewards and $R(s, a)$ is the true reward. In our experiments, we first train the agents with MADDPG (MA), M3DDPG (M3), and R-MADDPG (RM). Then we evaluate the quality of learned policies in a combination fashion, where each agent and adversary can be selected as the trained models from any of the aforementioned algorithms. We now demonstrate how these combinations lead to performance discrepancy in the environments with different levels of reward uncertainties. We report statistics that are averaged across 5 runs for cooperative navigation, and 25 runs for other scenarios where each agent or adversary is trained five times.

**Cooperative navigation.** Three agents learn to occupy all three landmarks as well as avoid collisions. Figure 1 and Figure 2 show the accumulated rewards and success rates during training, respectively. The figures indicate that when there is no reward uncertainty, the three models perform similarly and reach the same level of accumulated rewards and success rate (almost 1.0). At higher uncertainty levels, however, M3DDPG and MADDPG are difficult to learn good policies to occupy all landmarks, while R-MADDPG still reaches much higher success rates. Figure 3 provides results with additional metrics, where it shows that as the uncertainty level increases, R-MADDPG agents still manage to occupy most landmarks, therefore the sum of the minimum distance between each landmark and its closest agent is smaller and the number of occupied landmarks is larger.

**Keep-away.** In this single-agent and single-adversary scenario, we compare different models by the metric of average steps that the agent or adversary occupies the target landmark [13]. In Table 1, we

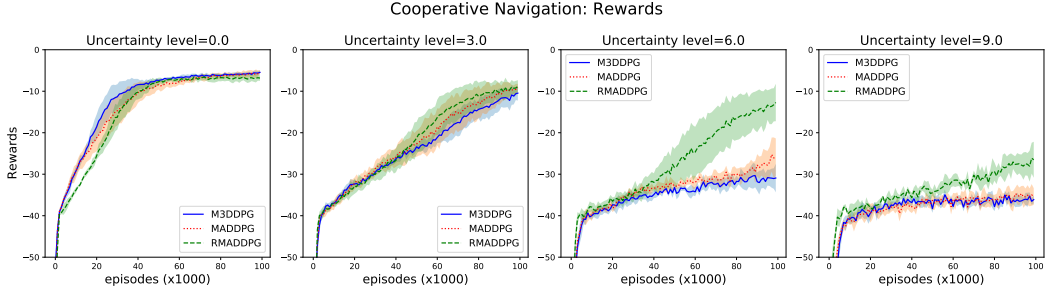

Figure 1: Cooperative navigation: accumulated rewards *vs* training episodes at different reward uncertainty levels. The shadow is the 95% confidence interval across five runs of each setting. Each point on the mean curve of the accumulated rewards is averaged across 1000 consecutive episodes.

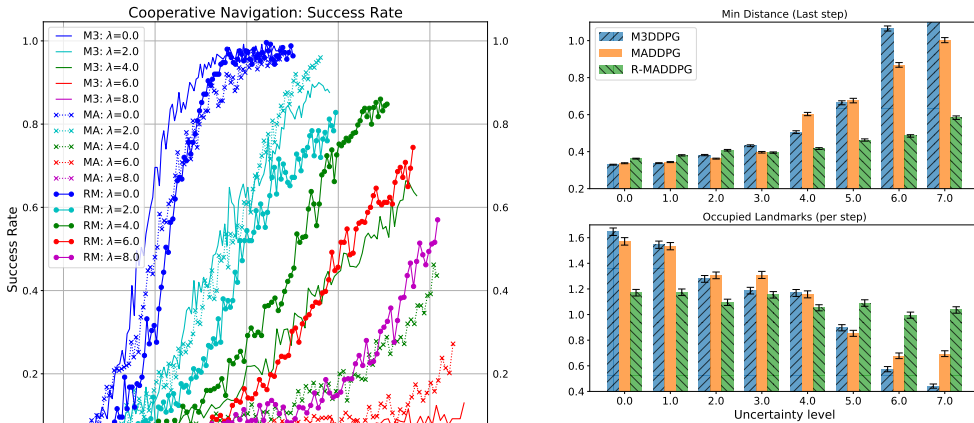

Figure 2: Success rate *vs* training time. An episode is successful if all landmarks are occupied by all agents. We only show the mean of each setting across five runs to avoid clutter.

Figure 3: *Top*: sum of min distance between each landmark and its closest agent at the last step of each episode (the smaller the better). *Bottom*: Average number of occupied landmarks per step (the larger the better). The error bars show 95% confidence interval across five runs of each setting.

Table 1: Keep-away: average steps for occupying the target per episode. We report the mean and 95% confidence interval from 25 model comparisons. Each model comparison evaluates 1000 episodes.

| Model | | $\lambda = 0$ | | $\lambda = 1.0$ | | $\lambda = 2.0$ | | $\lambda = 3.0$ | |
| Agent | Adversary | AG | ADV | AG | ADV | AG | ADV | AG | ADV |
|---|---|---|---|---|---|---|---|---|---|
| M3 | M3 | $16.65_{\pm1.49}$ | $12.22_{\pm2.07}$ | $9.70_{\pm2.96}$ | $5.84_{\pm2.42}$ | $4.26_{\pm2.27}$ | $2.94_{\pm1.88}$ | $1.43_{\pm1.49}$ | $1.69_{\pm1.44}$ |
| M3 | MA | $16.67_{\pm1.49}$ | $9.60_{\pm2.47}$ | $9.61_{\pm2.95}$ | $7.47_{\pm2.53}$ | $4.19_{\pm2.24}$ | $3.28_{\pm1.93}$ | $1.47_{\pm1.51}$ | $1.63_{\pm1.39}$ |
| M3 | RM | $16.68_{\pm1.49}$ | $8.65_{\pm2.45}$ | $9.53_{\pm2.94}$ | $7.99_{\pm2.55}$ | $4.17_{\pm2.23}$ | $3.34_{\pm2.00}$ | $1.38_{\pm1.44}$ | $3.1_{\pm1.92}$ |
| MA | M3 | $16.76_{\pm1.47}$ | $12.09_{\pm2.07}$ | $6.31_{\pm2.80}$ | $5.96_{\pm2.46}$ | $3.82_{\pm2.31}$ | $2.95_{\pm1.89}$ | $1.38_{\pm1.41}$ | $1.43_{\pm1.33}$ |
| MA | MA | $16.76_{\pm1.48}$ | $9.41_{\pm2.48}$ | $6.23_{\pm2.77}$ | $7.33_{\pm2.54}$ | $3.75_{\pm2.29}$ | $3.44_{\pm2.00}$ | $1.37_{\pm1.41}$ | $1.5_{\pm1.36}$ |
| MA | RM | $16.76_{\pm1.48}$ | $8.65_{\pm2.45}$ | $6.21_{\pm2.78}$ | $8.09_{\pm2.58}$ | $3.62_{\pm2.25}$ | $3.56_{\pm2.07}$ | $1.32_{\pm1.36}$ | $2.83_{\pm1.89}$ |
| RM | M3 | $4.9_{\pm2.50}$ | $9.95_{\pm2.44}$ | $10.19_{\pm3.06}$ | $6.19_{\pm2.46}$ | $7.18_{\pm2.79}$ | $3.4_{\pm2.01}$ | $5.32_{\pm2.64}$ | $1.61_{\pm1.37}$ |
| RM | MA | $5.05_{\pm2.56}$ | $7.37_{\pm2.55}$ | $10.15_{\pm3.06}$ | $8.02_{\pm2.54}$ | $7.21_{\pm2.78}$ | $3.69_{\pm2.04}$ | $5.35_{\pm2.64}$ | $1.94_{\pm1.54}$ |
| RM | RM | $5.22_{\pm2.61}$ | $7.21_{\pm2.51}$ | $10.02_{\pm3.06}$ | $8.56_{\pm2.52}$ | $7.27_{\pm2.78}$ | $4.12_{\pm2.20}$ | $5.26_{\pm2.61}$ | $3.73_{\pm2.06}$ |

present the results of every possible combination for completeness, although we are only interested in fixing the agent or adversary as the one trained from R-MADDPG and varying the opponent among R-MADDPG, M3DDPG, and MADDPG. More specifically, we evaluate the quality of policies learned from each algorithm when their opponents act in a robust way. From the table, we can observe that when fixing the agent or adversary as R-MADDPG, the M3DDPG opponent performs

Table 2: Physical deception: success rates of agents and adversary, and minimum distance of agents from the non-target landmark. The results are averaged across 25 runs.

| Model | | $\lambda = 0$ | | | $\lambda = 1.0$ | | | $\lambda = 2.0$ | | | $\lambda = 3.0$ | | |
| Agents | Adversary | AG/ADV succ rate, AG dist to non-target | | | AG/ADV succ rate, AG dist to non-target | | | AG/ADV succ rate, AG dist to non-target | | | AG/ADV succ rate, AG dist to non-target | | |
|---|---|---|---|---|---|---|---|---|---|---|---|---|---|
| MA | MA | 0.87 | 0.40 | 0.25 | 0.76 | 0.49 | 0.45 | 0.68 | 0.54 | 0.64 | 0.61 | 0.63 | 0.89 |
| MA | RM | 0.87 | 0.47 | 0.25 | 0.76 | 0.52 | 0.45 | 0.68 | 0.53 | 0.64 | 0.61 | 0.50 | 0.89 |
| RM | MA | 0.83 | 0.42 | 0.26 | 0.77 | 0.52 | 0.41 | 0.86 | 0.66 | 0.61 | 0.72 | 0.56 | 0.54 |
| RM | RM | 0.83 | 0.48 | 0.27 | 0.77 | 0.55 | 0.41 | 0.86 | 0.72 | 0.61 | 0.73 | 0.57 | 0.54 |

Table 3: Predator-prey: total number of prey touches by predators per episode. For prey, the smaller the better. For predators, the larger the better. The results are averaged across 25 runs.

| Model | | Uncertainty level ($\lambda$) | | | |
| Prey (Agent) | Predators (Adversaries) | 0 | 1.0 | 2.0 | 3.0 |
|---|---|---|---|---|---|
| MADDPG | MADDPG | $2.31_{\pm 1.31}$ | $2.38_{\pm 1.41}$ | $2.85_{\pm 1.51}$ | $3.20_{\pm 1.72}$ |
| R-MADDPG | MADDPG | $2.15_{\pm 1.22}$ | $1.61_{\pm 1.12}$ | $2.42_{\pm 1.37}$ | $2.78_{\pm 1.56}$ |
| MADDPG | R-MADDPG | $3.40_{\pm 1.68}$ | $3.82_{\pm 1.81}$ | $3.19_{\pm 1.64}$ | $4.64_{\pm 2.04}$ |
| R-MADDPG | R-MADDPG | $2.66_{\pm 1.42}$ | $2.37_{\pm 1.36}$ | $2.69_{\pm 1.53}$ | $3.58_{\pm 1.78}$ |

better than the other two when there is no uncertainty ($\lambda = 0$) in the environment. As uncertainty level increases, R-MADDPG consistently outperforms MADDPG and M3DDPG. This is because the agent or adversary trained with MADDPG and M3DDPG starts to be confusing, while the R-MADDPG models can still correctly infer the goal and occupy it.

**Physical deception.** We generate two agents and one adversary, and use different metrics to evaluate the performance in this experiment. We first report the success rates of agents and adversary that occupy the target landmark at the final step. In addition, to evaluate the deception strategy of agents, we also compute the minimum distance of agents from the non-target landmark, which indicates how well the agents spread out and cover all landmarks. The results are provided in Table 2. When fixing the adversary as R-MADDPG models (*ref.* the second and fourth rows of the table), the R-MADDPG agents perform better than the MADDPG agents for $\lambda > 0$ in both the success rates and distance to the non-target landmark (the smaller the better). Moreover, one can get similar observation when the two agents are fixed to be R-MADDPG models with different adversaries (*ref.* last two rows).

**Predator-prey.** We create three predators and two obstacle landmarks, and evaluate the policies by the average number of the lone prey hit by the predators per episode. The results are presented in Table 3. Surprisingly, in this case, R-MADDPG models consistently work better than MADDPG models with or without model uncertainty in the environment. When fixing the prey as R-MADDPG, R-MADDPG predators always hit the prey more than the MADDPG predators do for every $\lambda$ value. Similarly, when fixing the predators as R-MADDPG, the R-MADDPG prey is able to avoid being caught by predators better than the MADDPG prey. We conjecture that when the environment is certain, the nature agent in R-MADDPG fits well to the reward function in this experiment, hence training with R-MADDPG and MADDPG performs homogeneously.

## 5 Concluding Remarks

In this work, we have advocated the use of robust Markov games to capture the model uncertainty in MARL problems, motivated by the sim-to-real gap in the autonomous-car racing application [17]. By viewing the uncertainty as the decision made by an implicit player, we then introduce the nature agent to model the uncertainty, who always plays against each agent by selecting the worst-case data at every state. To find the solution concept of robust Nash equilibrium in this model, we first develop a Q-learning algorithm with convergence guarantees under certain conditions. In addition, we have also proposed a multi-agent actor-critic method, i.e., Robust-MADDPG, to incorporate function approximation and handle large state-action spaces. Our experiments in multiple benchmark environments have shown the effectiveness of Robust-MADDPG in addressing the uncertainty in MARL, outperforming several MARL methods with no robustness concerns. As future work, we plan to apply our method to other MARL scenarios with model uncertainty, and evaluate its sim-to-real performance in practical robotics applications, e.g., the multi-car racing platform [17].

## Broader Impact

We believe that researchers of multi-agent reinforcement learning (MARL) and robust RL would benefit from this work, as we have explored one possibility to systematically handle model-uncertainty in MARL. In particular, prior to this work, though a common issue in practice, it is unknown yet how to deal with the uncertainties of the model in MARL, to address the sim-to-real gap in MARL. We have made an initial attempt to address this issue, under a theoretical framework of robust Markov games. In light of the ubiquity of RL on multi-agent systems, especially those safety-critical ones, e.g., autonomous-driving cars, robots, unmanned aerial vehicles, *safe MARL*, the broader topic that our work belongs to, would be of paramount importance, and would eventually push forward the application of MARL on practical systems. Our work will hopefully bring the topic of safe MARL into researchers' attention, and open up several interesting and challenging future research directions along the line. We do not believe that our research will cause any ethical issue, or put anyone at any disadvantage.

## Acknowledgments and Disclosure of Funding

The research of K.Z. and T.B. was supported in part by the US Army Research Laboratory (ARL) Cooperative Agreement W911NF-17-2-0196, and in part by the Office of Naval Research (ONR) MURI Grant N00014-16-1-2710. The authors would also like to thank the anonymous reviewers for the valuable comments.

## Footnotes

[2] For simplicity, we omit the superscript $i$ since the index $i$ can be identified by the parameter used.

[3] Note that the derivation below can be easily generalized to the setting that the initial state is randomly drawn from some distribution.

[4] For notational simplicity, we omit the parameter that some function takes gradient with respect to, if the function takes gradient with respect to the full parameter, e.g., we write $\nabla_{\theta^{0,i}} \pi_{\theta^{0,i}}(s)[a]$ as $\nabla \pi_{\theta^{0,i}}(s)[a]$, $\nabla_{\theta^i} \log \pi_{\theta^i}(a^i \,|\, s)$ as $\nabla \log \pi_{\theta^i}(a^i \,|\, s)$.

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
