[Supplementary Material · supple.pdf]

# Supplementary Material for "Robust Multi-Agent Reinforcement Learning with Model Uncertainty"

## A   Supplementary Proofs

### A.1   Proof of Proposition 2.2

The result is a direct application of Theorem 4 in [23]. The minor difference is that the reward uncertainty set $\bar{\mathcal{R}}_s^i$ here is a vector lying in $\mathbb{R}^{|\mathcal{A}|}$, which might be different across agents. While in [23], the cost uncertainty set $C_s$ is a vector of dimension $|\mathcal{N}||\mathcal{A}|$. Thus, each element in $C_s$ therein should be equivalent to the concatenated vector $((\bar{\mathcal{R}}_s^1)^\top, (\bar{\mathcal{R}}_s^2)^\top, \cdots, (\bar{\mathcal{R}}_s^N)^\top)^\top$, which also lies in a compact set since each subvector lies in a compact set. This enables the application of [23, Theorem 4] to obtain the existence result. $\square$

### A.2   Convergence of Q-learning Under Certain Conditions

We now provide some theoretical justifications for the convergence of the Q-learning update proposed in §3.1. In the following, we rely on the assumptions based on the final version of the formulation by [25], where the authors resolve issues with their initial formulation. While convergence to Nash equilibrium under these assumptions are guaranteed, we understand these assumptions are a bit restrictive for practical applications. Additional discussion on the shortcomings surrounding the use of convergence to Nash equilibrium can be found in literature. Hence, we view our result as a proof-of-concept for justifying the value-based/Q-learning update. Indeed, developing provable convergent Q-learning for general-sum Markov games without restrictive assumptions remains open, and is still worth further investigation. Hence, we have been motivated to develop an actor-critic algorithm later, which incorporates function approximation to handle more practical cases.

Recall that the Q-value function at the RMPNE satisfies the following Bellman equation:

$$\bar{Q}_*^i\big(s, a, \bar{R}^i(s,a)\big) := \bar{R}^i(s,a) + \gamma \sum_{s' \in \mathcal{S}} P(s' \mid s, a) \sum_{a'} \bigg( \prod_{j=1}^N \pi_*^j(a'^j \mid s') \bigg) \bar{Q}_*^i\big(s', a', \bar{R}_*^i(s',a')\big). \tag{A.1}$$

The Q-learning update can be written as

$$\bar{Q}_{t+1}^i(s_t, a_t, \bar{R}_t^i) := (1 - \alpha_t) \cdot \bar{Q}_t^i(s_t, a_t, \bar{R}_t^i) + \alpha_t \cdot \Big[ \bar{R}_t^i + \gamma \sum_{a_{t+1}} \pi_{*,t}(a_{t+1} \mid s_{t+1}) \cdot$$
$$\bar{Q}_t^i(s_{t+1}, a_{t+1}, \bar{R}_{t+1}^i) \Big]. \tag{A.2}$$

We consider the setting with two agents for simplicity, and make the following assumptions, motivated from [25].

**Assumption 4.1.** Every state and action have been visited infinitely often.

**Assumption 4.2.** The learning rate $\alpha_t$ satisfies the following conditions:

- $0 \le \alpha_t < 1$, $\sum_{t \ge 0} \alpha_t = \infty$, and $\sum_{t \ge 0} \alpha_t^2 < \infty$,

- $\alpha_t(s, a^1, a^2, \bar{R}^i(s,a)) = 0$ if $(s, a^1, a^2, \bar{R}^i(s,a)) \ne (s_t, a_t^1, a_t^2, \bar{R}_t^i)$.

**Assumption 4.3.** Define $\bar{Q}_t^i(s) = [\bar{Q}_t^i(s, a^1, a^2, \bar{R}^i(s,a))]_{a^1 \in \mathcal{A}^1, a^2 \in \mathcal{A}^2, \bar{R}_s^i \in \bar{\mathcal{R}}_s^i}$ to be the estimates of Q-value functions at iteration $t$ of (A.2), and define the *stage* RMPNE for $(\bar{Q}_t^1(s), \bar{Q}_t^2(s))$ as the tuple of policies $\big(\{\pi_*^{0,i}(s)\}_{i \in \mathcal{N}}, \pi_*^1(\cdot \mid s), \pi_*^2(\cdot \mid s)\big)$ that is obtained from

$$\big(\pi_*^i(\cdot \mid s), \pi_*^{0,i}(s)\big) \in \underset{\pi^i(\cdot \mid s)}{\arg\max} \min_{\pi^{0,i}(s)} \sum_{a \in \mathcal{A}} \pi^i(a^i \mid s) \pi_*^{-i}(a^{-i} \mid s) \bar{Q}_t^i\big(s, a, \pi^{0,i}(s)[a]\big), \tag{A.3}$$

where $-i = 1$ if $i = 2$, and $-i = 2$ if $i = 1$. Moreover, the stage equilibrium policy tuple satisfies one of the following properties:

- The equilibrium policy tuple is global optimum, i.e., for any $\pi^i(\cdot \mid s) \in \Delta(\mathcal{A}^i)$ with $i = 1, 2$ and $\pi^{0,i}(s) \in \bar{\mathcal{R}}_s^i$,

$$\sum_{a \in \mathcal{A}} \pi_*^i(a^i \mid s)\pi_*^{-i}(a^{-i} \mid s)\bar{Q}_t^i\big(s, a, \pi_*^{0,i}(s)[a]\big) \geq \sum_{a \in \mathcal{A}} \pi^i(a^i \mid s)\pi^{-i}(a^{-i} \mid s)\bar{Q}_t^i\big(s, a, \pi^{0,i}(s)[a]\big).$$

- One agent receives a higher payoff when the other agent deviates from the equilibrium policy tuple, i.e., for any $\pi^i(\cdot \mid s) \in \Delta(\mathcal{A}^i)$ and $\pi^{0,i}(s) \in \bar{\mathcal{R}}_s^i$ with $i = 1, 2$

$$\sum_{a \in \mathcal{A}} \pi_*^1(a^1 \mid s)\pi_*^2(a^2 \mid s)\bar{Q}_t^1\big(s, a, \pi^{0,1}(s)[a]\big) \leq \sum_{a \in \mathcal{A}} \pi_*^1(a^1 \mid s)\pi^2(a^2 \mid s)\bar{Q}_t^1\big(s, a, \pi^{0,1}(s)[a]\big),$$

$$\sum_{a \in \mathcal{A}} \pi_*^2(a^2 \mid s)\pi_*^1(a^1 \mid s)\bar{Q}_t^2\big(s, a, \pi^{0,2}(s)[a]\big) \leq \sum_{a \in \mathcal{A}} \pi_*^2(a^2 \mid s)\pi^1(a^1 \mid s)\bar{Q}_t^2\big(s, a, \pi^{0,2}(s)[a]\big).$$

With Assumptions 4.1-4.3, we can prove the convergence of Q-learning in the following theorem.

**Theorem 4.4.** Under Assumptions 4.1-4.3, the sequence $\{(\bar{Q}_t^1, \bar{Q}_t^2)\}$ obtained from (A.2) converges to $(\bar{Q}_*^1, \bar{Q}_*^2)$, which are the optimal Q-value functions that solve the Bellman equation (A.1), namely, the robust Markov perfect Nash equilibrium Q-value.

*Proof.* Define the operator

$$\mathcal{P}_t^i \bar{Q}^i(s) = \bar{R}_t^i + \gamma \sum_{a \in \mathcal{A}} \pi_*^i(a^i \mid s)\pi_*^{-i}(a^{-i} \mid s)\bar{Q}^i\big(s, a, \pi_*^{0,i}(s)[a]\big), \tag{A.4}$$

for $i = 1, 2$, where $\big(\{\pi_*^{0,i}(s)\}_{i=1,2}, \pi_*^1(\cdot \mid s), \pi_*^2(\cdot \mid s)\big)$ is the tuple of equilibrium policies for $(\bar{Q}^1(s), \bar{Q}^2(s))$ obtained from (A.3). We first show that $\mathcal{P}_t = (\mathcal{P}_t^1, \mathcal{P}_t^2)$ is a contraction mapping.

**Lemma A.1.** Let $\mathcal{P}_t = (\mathcal{P}_t^1, \mathcal{P}_t^2)$ where $\mathcal{P}_t^i$ is defined in (A.4). Then $\mathcal{P}_t$ is a contraction mapping under Assumption 4.3.

*Proof.* Consider two pairs of Q-values at state $s$ denoted by $(\bar{Q}^1(s), \bar{Q}^2(s))$ and $(\hat{Q}^1(s), \hat{Q}^2(s))$, respectively, whose equilibrium tuples are denoted by

$$\big(\{\pi_*^{0,i}(s)\}_{i=1,2}, \pi_*^1(\cdot \mid s), \pi_*^2(\cdot \mid s)\big), \quad \text{and} \quad \big(\{\hat{\pi}_*^{0,i}(s)\}_{i=1,2}, \hat{\pi}_*^1(\cdot \mid s), \hat{\pi}_*^2(\cdot \mid s)\big).$$

To show the contraction property, we consider the following two cases.

**Case** 1: $\mathcal{P}_t^i \bar{Q}^i(s) \geq \mathcal{P}_t^i \hat{Q}^i(s)$. Then under the first property of Assumption 4.3, i.e., the global optimality of the equilibrium, we have

$$0 \leq \mathcal{P}_t^1 \bar{Q}^1(s) - \mathcal{P}_t^1 \hat{Q}^1(s)$$

$$= \gamma \left[ \sum_{a \in \mathcal{A}} \pi_*^i(a^i \mid s)\pi_*^{-i}(a^{-i} \mid s)\bar{Q}^i\big(s, a, \pi_*^{0,i}(s)[a]\big) - \sum_{a \in \mathcal{A}} \hat{\pi}_*^i(a^i \mid s)\hat{\pi}_*^{-i}(a^{-i} \mid s)\hat{Q}^i\big(s, a, \hat{\pi}_*^{0,i}(s)[a]\big) \right]$$

$$\leq \gamma \left[ \sum_{a \in \mathcal{A}} \pi_*^i(a^i \mid s)\pi_*^{-i}(a^{-i} \mid s)\bar{Q}^i\big(s, a, \pi_*^{0,i}(s)[a]\big) - \sum_{a \in \mathcal{A}} \pi_*^i(a^i \mid s)\pi_*^{-i}(a^{-i} \mid s)\hat{Q}^i\big(s, a, \pi_*^{0,i}(s)[a]\big) \right]$$

$$\leq \gamma \max_{a^1, a^2, \bar{R}_s^i} \left| \bar{Q}^i(s, a^1, a^2, \bar{R}^i(s, a)) - \hat{Q}^i(s, a^1, a^2, \bar{R}^i(s, a)) \right| = \gamma \|\bar{Q}^i(s) - \hat{Q}^i(s)\|_\infty, \tag{A.5}$$

where the second inequality uses this property. Furthermore, under the second property of Assumption 4.3, we can derive

$$0 \leq \mathcal{P}_t^1 \bar{Q}^1(s) - \mathcal{P}_t^1 \hat{Q}^1(s)$$

$$= \gamma \left[ \sum_{a \in \mathcal{A}} \pi_*^i(a^i \mid s)\pi_*^{-i}(a^{-i} \mid s)\bar{Q}^i\big(s, a, \pi_*^{0,i}(s)[a]\big) - \sum_{a \in \mathcal{A}} \hat{\pi}_*^i(a^i \mid s)\hat{\pi}_*^{-i}(a^{-i} \mid s)\hat{Q}^i\big(s, a, \hat{\pi}_*^{0,i}(s)[a]\big) \right]$$

$$\leq \gamma \left[ \sum_{a \in \mathcal{A}} \pi_*^i(a^i \mid s)\pi_*^{-i}(a^{-i} \mid s)\bar{Q}^i\big(s, a, \pi_*^{0,i}(s)[a]\big) - \sum_{a \in \mathcal{A}} \pi_*^i(a^i \mid s)\hat{\pi}_*^{-i}(a^{-i} \mid s)\hat{Q}^i\big(s, a, \pi_*^{\prime 0,i}(s)[a]\big) \right]$$

$$\leq \gamma \left[ \sum_{a \in \mathcal{A}} \pi_*^i(a^i \mid s)\pi_*^{-i}(a^{-i} \mid s)\bar{Q}^i\big(s, a, \pi_*^{\prime 0,i}(s)[a]\big) - \sum_{a \in \mathcal{A}} \pi_*^i(a^i \mid s)\hat{\pi}_*^{-i}(a^{-i} \mid s)\hat{Q}^i\big(s, a, \pi_*^{\prime 0,i}(s)[a]\big) \right]$$

$$\leq \gamma \left[ \sum_{a \in \mathcal{A}} \pi_*^i(a^i \mid s)\hat{\pi}_*^{-i}(a^{-i} \mid s)\bar{Q}^i\big(s, a, \pi_*^{\prime 0,i}(s)[a]\big) - \sum_{a \in \mathcal{A}} \pi_*^i(a^i \mid s)\hat{\pi}_*^{-i}(a^{-i} \mid s)\hat{Q}^i\big(s, a, \pi_*^{\prime 0,i}(s)[a]\big) \right]$$

$$\leq \gamma \|\bar{Q}^i(s) - \hat{Q}^i(s)\|_\infty, \tag{A.6}$$

where the second inequality uses the definition of the equilibrium, with $\pi_*^{\prime 0,i}(s)$ denoting the minimizer of $\hat{Q}^i(s)$ corresponding to $\pi_*^i$; the third inequality is due to that for fixed $\pi_*^1(\cdot \mid s)$ and $\pi_*^2(\cdot \mid s)$, $\pi_*^{0,i}(s)$ is the minimizer; the fourth inequality uses the second property of Assumption 4.3; the last inequality follows by the definition of $\| \cdot \|_\infty$-norm. Both (A.5) and (A.6) lead to a $\gamma$-contraction in $\| \cdot \|_\infty$ norm.

**Case** 2: $\mathcal{P}_t^i \bar{Q}^i(s) \leq \mathcal{P}_t^i \hat{Q}^i(s)$. Similar arguments apply for this case, which are omitted here for brevity.

Note that for both cases, the $\gamma$-contraction result holds for any $s \in \mathcal{S}$, which completes the proof. $\square$

Let $\bar{Q}^i = [\bar{Q}^i(s)]_{s \in \mathcal{S}}$ for any $\bar{Q}^i$. Then Lemma A.1 means that $\|\mathcal{P}_t^i \bar{Q}^i - \mathcal{P}_t^i \bar{Q}_*^i\|_\infty \leq \gamma \|\bar{Q}^i - \bar{Q}_*^i\|_\infty$ for any $\bar{Q}^i$. In sum, the operator $\mathcal{P}_t$ satisfies both conditions: i) it is a contraction mapping; ii) $\bar{Q}_*^i$ is a fixed point of $\bar{Q}_*^i = \mathbb{E}(\mathcal{P}_t^i \bar{Q}_*^i)$ for $i = 1, 2$. By Lemma 8 in [25] (also Corollary 5 in [49]), we know that $\{(\bar{Q}_t^1, \bar{Q}_t^2)\}$ converges to $\{(\bar{Q}_*^1, \bar{Q}_*^2)\}$, which concludes the proof. $\square$

The convergence to the Q-value at the equilibrium further proves the convergence to the equilibrium policies, which are obtained based on the Q-value estimate $\bar{Q}_t = (\bar{Q}_t^1, \cdots, \bar{Q}_t^N)$ at iteration $t$, by solving

$$\left(\pi_{*,t}^i(\cdot \mid s), \pi_{*,t}^{0,i}(s)\right) \in \underset{\pi^i(\cdot \mid s)}{\operatorname{argmax}} \min_{\pi^{0,i}(s)} \sum_{a \in \mathcal{A}} \pi^i(a^i \mid s) \prod_{j \neq i} \pi_*^j(a^j \mid s) \bar{Q}_t^i\left(s, a, \pi^{0,i}(s)[a]\right),$$

where $\pi^{0,i}(s)[a]$ is the $a$-th element of the output vector $\pi^{0,i}(s)[a]$.

# B Policy Gradient Theorem in Robust MARL

We now prove the policy gradient theorem in robust MARL, as previously stated in Lemma 3.1. For completeness, we here derive a more general version of the theorem, allowing both the reward function and transition probability distribution being parametrized. We first introduce $\theta^{0,0}$ to be the parameter of the transition model $P_{\theta^{0,0}}$, making the joint policy to be $\widetilde{\pi}_\theta := (\pi_{\theta^0}, \pi_{\theta^1}, \cdots, \pi_{\theta^N})$, with parameter $\theta = (\theta^0, \theta^1, \cdots, \theta^N)$ and $\theta^0 := (\theta^{0,0}, \theta^{0,1}, \cdots, \theta^{0,N})$. We can then define the return objective of each agent $i$ under the joint policy $\widetilde{\pi}_\theta$ as $J^i(\theta) := \bar{V}_{\widetilde{\pi}_\theta}^i(s')$, where the value function $\bar{V}_{\widetilde{\pi}_\theta}^i$ satisfies

$$\bar{V}_{\widetilde{\pi}_\theta}^i(s) = \sum_{a \in \mathcal{A}} \prod_{j=1}^N \pi_{\theta^j}(a^j \mid s) \left(\pi_{\theta^{0,i}}(s)[a] + \gamma \sum_{s' \in \mathcal{S}} P_{\theta^{0,0}}(s' \mid s, a) \bar{V}_{\widetilde{\pi}_\theta}^i(s')\right), \quad \text{(B.1)}$$

in contrast to (3.4) without transition parametrization. Similarly one can define the Q-value function under the joint policy $\widetilde{\pi}_\theta$, denoted by $\bar{Q}_{\widetilde{\pi}_\theta}^i$. We then state the complete version as follows.

**Lemma B.1** (Policy Gradient Theorem in Robust MARL). For each agent $i = 1, \cdots, N$, the policy gradient of the objective $J^i(\theta)$ with respect to the parameter $\theta$ has the following form:

$$\nabla_{\theta^i} J^i(\theta) = \mathbb{E}_{s \sim \rho_{\pi_\theta}^{s_0}, a \sim \pi_\theta(\cdot \mid s)}\left[\nabla \log \pi_{\theta^i}(a^i \mid s) \bar{Q}_{\widetilde{\pi}_\theta}^i(s, a)\right], \quad \text{(B.2)}$$

$$\nabla_{\theta^{0,i}} J^i(\theta) = \mathbb{E}_{s \sim \rho_{\pi_\theta}^{s_0}, a \sim \pi_\theta(\cdot \mid s)}\left[\nabla \pi_{\theta^{0,i}}(s)[a]\right], \quad \text{(B.3)}$$

$$\nabla_{\theta^{0,0}} J^i(\theta) = \mathbb{E}_{s \sim \rho_{\pi_\theta}^{s_0}, a \sim \pi_\theta(\cdot \mid s), s' \sim P_{\theta^{0,0}}(\cdot \mid s, a)}\left[\gamma \nabla \log P_{\theta^{0,0}}(s' \mid s, a) \cdot \bar{V}_{\widetilde{\pi}_\theta}^i(s')\right], \quad \text{(B.4)}$$

where $\pi_\theta(a \mid s) := \prod_{j=1}^N \pi_{\theta^j}(a^j \mid s)$, $\rho_{\pi_\theta}^{s_0}(s) := \sum_{t=0}^\infty \gamma^t \cdot Pr(s_0 \to s, t, \pi_\theta)$ is the discounted state visitation measure under joint policy $\pi_\theta$ with state starting from $s_0$, with $Pr(s \to s', t, \pi_\theta)$ denoting the probability of transitioning from $s$ to $s'$ under joint policy $\pi_\theta$ with $t$-steps, and $\pi_{\theta^{0,i}}(s)[a]$ is the $a$-th element of the output of $\pi_{\theta^{0,i}}(s)$.

*Proof.* Note that $J^i(\theta)$ can be viewed as the standard value in Markov games with reward function $R^i(s, a) = \pi_{\theta^{0,i}}(s)[a]$. Thus, the form of (B.2) follows by the derivation in either [13, Eq. (4)] or [15, Theorem 3.1].

Moreover, taking gradient with respect to $\theta^{0,i}$ for $i \in \mathcal{N}$ on both sides of (B.1) yields

$$\nabla_{\theta^{0,i}} \bar{V}^i_{\tilde{\pi}_\theta}(s) = \sum_{a \in \mathcal{A}} \pi_\theta(a \mid s) \left( \nabla \pi_{\theta^{0,i}}(s)[a] + \gamma \sum_{s' \in \mathcal{S}} P(s' \mid s, a) \cdot \nabla_{\theta^{0,i}} \bar{V}^i_{\tilde{\pi}_\theta}(s') \right)$$

$$= \sum_{a \in \mathcal{A}} \pi_\theta(a \mid s) \Bigg[ \nabla \pi_{\theta^{0,i}}(s)[a] + \gamma \sum_{s' \in \mathcal{S}} P(s' \mid s, a) \cdot \sum_{a' \in \mathcal{A}} \pi_\theta(a' \mid s')$$

$$\left( \nabla \pi_{\theta^{0,i}}(s')[a'] + \gamma \sum_{s'' \in \mathcal{S}} P(s'' \mid s', a') \cdot \nabla_{\theta^{0,i}} \bar{V}^i_{\tilde{\pi}_\theta}(s'') \right) \Bigg]$$

$$= \mathbb{E}_{a \sim \pi_\theta(\cdot \mid s)} \left[ \nabla \pi_{\theta^{0,i}}(s)[a] \right] + \gamma \sum_{s' \in \mathcal{S}} Pr(s \to s', 1, \pi_\theta) \mathbb{E}_{a' \sim \pi_\theta(\cdot \mid s')} \left[ \nabla \pi_{\theta^{0,i}}(s')[a'] \right]$$

$$+ \gamma^2 \sum_{s'' \in \mathcal{S}} Pr(s \to s'', 2, \pi_\theta) \cdot \nabla_{\theta^{0,i}} \bar{V}^i_{\tilde{\pi}_\theta}(s''), \tag{B.5}$$

where the second equation follows by unrolling $\nabla_{\theta^{0,i}} \bar{V}^i_{\tilde{\pi}_\theta}(s')$. By keeping unrolling (B.5), we have

$$\nabla_{\theta^{0,i}} \bar{V}^i_{\tilde{\pi}_\theta}(s) = \sum_{s' \in \mathcal{S}} \sum_{t=0}^{\infty} \gamma^t Pr(s \to s', t, \pi_\theta) \cdot \mathbb{E}_{a' \sim \pi_\theta(\cdot \mid s')} \left[ \nabla \pi_{\theta^{0,i}}(s')[a'] \right]$$

$$= \sum_{s' \in \mathcal{S}} \rho^s_{\pi_\theta}(s') \cdot \mathbb{E}_{a' \sim \pi_\theta(\cdot \mid s')} \left[ \nabla \pi_{\theta^{0,i}}(s')[a'] \right], \tag{B.6}$$

which implies the formula in (B.3).

Finally, taking gradient with respect to $\theta^{0,0}$ on both sides of (B.1), we have

$$\nabla_{\theta^{0,0}} \bar{V}^i_{\tilde{\pi}_\theta}(s) = \gamma \sum_{a \in \mathcal{A}} \pi_\theta(a \mid s) \sum_{s' \in \mathcal{S}} \left( \nabla_{\theta^{0,0}} P_{\theta^{0,0}}(s' \mid s, a) \cdot \bar{V}^i_{\tilde{\pi}_\theta}(s') + P_{\theta^{0,0}}(s' \mid s, a) \cdot \nabla_{\theta^{0,0}} \bar{V}^i_{\tilde{\pi}_\theta}(s') \right)$$

$$= \sum_{a \in \mathcal{A}} \pi_\theta(a \mid s) \Bigg[ \gamma \sum_{s' \in \mathcal{S}} \nabla_{\theta^{0,0}} P_{\theta^{0,0}}(s' \mid s, a) \cdot \bar{V}^i_{\tilde{\pi}_\theta}(s') + \gamma \sum_{s' \in \mathcal{S}} P_{\theta^{0,0}}(s' \mid s, a) \cdot \nabla_{\theta^{0,0}} \bar{V}^i_{\tilde{\pi}_\theta}(s') \Bigg]$$

$$= \sum_{a \in \mathcal{A}} \pi_\theta(a \mid s) \Bigg[ \gamma \sum_{s' \in \mathcal{S}} \nabla_{\theta^{0,0}} P_{\theta^{0,0}}(s' \mid s, a) \cdot \bar{V}^i_{\tilde{\pi}_\theta}(s') + \gamma \sum_{s' \in \mathcal{S}} P_{\theta^{0,0}}(s' \mid s, a) \cdot \sum_{a' \in \mathcal{A}} \pi_\theta(a' \mid s')$$

$$\left( \gamma \sum_{s'' \in \mathcal{S}} \nabla_{\theta^{0,0}} P_{\theta^{0,0}}(s'' \mid s', a') \cdot \bar{V}^i_{\tilde{\pi}_\theta}(s'') + \gamma \sum_{s'' \in \mathcal{S}} P_{\theta^{0,0}}(s'' \mid s', a') \cdot \nabla_{\theta^{0,0}} \bar{V}^i_{\tilde{\pi}_\theta}(s'') \right) \Bigg]$$

$$= \mathbb{E}_{a \sim \pi_\theta(\cdot \mid s), s' \sim P_{\theta^{0,0}}(\cdot \mid s, a)} \left[ \gamma \nabla \log P_{\theta^{0,0}}(s' \mid s, a) \cdot \bar{V}^i_{\tilde{\pi}_\theta}(s') \right]$$

$$+ \gamma \sum_{s' \in \mathcal{S}} Pr(s \to s', 1, \pi_\theta) \cdot \mathbb{E}_{a' \sim \pi_\theta(\cdot \mid s'), s'' \sim P_{\theta^{0,0}}(\cdot \mid s', a')} \left[ \gamma \nabla \log P_{\theta^{0,0}}(s'' \mid s', a') \cdot \bar{V}^i_{\tilde{\pi}_\theta}(s'') \right]$$

$$+ \gamma^2 \sum_{s'' \in \mathcal{S}} Pr(s \to s'', 2, \pi_\theta) \cdot \nabla_{\theta^{0,0}} \bar{V}^i_{\tilde{\pi}_\theta}(s''). \tag{B.7}$$

By keeping unrolling (B.7), we have

$$\nabla_{\theta^{0,0}} \bar{V}^i_{\tilde{\pi}_\theta}(s)$$

$$= \sum_{s' \in \mathcal{S}} \sum_{t=0}^{\infty} \gamma^t Pr(s \to s', t, \pi_\theta) \cdot \mathbb{E}_{a' \sim \pi_\theta(\cdot \mid s'), s'' \sim P_{\theta^{0,0}}(\cdot \mid s', a')} \left[ \gamma \nabla \log P_{\theta^{0,0}}(s'' \mid s', a') \cdot \bar{V}^i_{\tilde{\pi}_\theta}(s'') \right]$$

$$= \sum_{s' \in \mathcal{S}} \rho^s_{\pi_\theta}(s') \cdot \mathbb{E}_{a' \sim \pi_\theta(\cdot \mid s'), s'' \sim P_{\theta^{0,0}}(\cdot \mid s', a')} \left[ \gamma \nabla \log P_{\theta^{0,0}}(s'' \mid s', a') \cdot \bar{V}^i_{\tilde{\pi}_\theta}(s'') \right], \tag{B.8}$$

which implies the policy gradient with respect to $\theta^{0,0}$ and completes the proof. $\qquad\square$

More specifically, for simplicity, suppose the uncertainty of the transition $P_{\theta^{0,0}}(\cdot \mid s, a)$ is parameterized as that for any $(s, a)$,

$$P_{\theta^{0,0}}(\cdot \mid s, a) = \mu \cdot P^{\texttt{pert}}_{\theta^{0,0}}(\cdot \mid s, a) + (1 - \mu) \cdot P(\cdot \mid s, a),$$

---

**Algorithm 1 Actor-Critic for Robust Multi-Agent RL (Robust-MADDPG):**

---

1: Initialization of Q-value parameters $\{\omega_0^i\}_{i\in\mathcal{N}}$, and policy parameters $\{\theta_0^i\}_{i\in\mathcal{N}}$ and $\theta_0^0 := \{\theta_0^{0,i}\}_{i\in\mathcal{N}}$.
2: **for** episode $= 1$ to $M$ **do**
3:     Receive an initial state $s$
4:     **for** $t = 1, \cdots T$ **do**
5:         For each agent $i$, sample action $a^i \sim \pi_{\theta_t^i}$ with current policy $\pi_{\theta_t^i}$
6:         Execute joint $a = (a^1, \cdots, a^N)$, and observe new state $s'$
7:         Each agent $i$ also receives a reward with uncertainty $\bar{r}^i$
8:         Store $(s, a, \bar{r}^i, s')$ for each $i$ in replay buffer $\mathcal{D}$, let $s' \leftarrow s$
9:         **for** agent $i = 1$ to $N$ **do**
10:           Sample a random mini-batch of $S$ samples of $(s_t, a_t, \bar{r}_t^i, s_{t+1})$ from $\mathcal{D}$
11:           Set

$$y_t = \pi_{\theta'^{0,i}}(s_t)[a_t] + \gamma \bar{Q}_{\omega'^i}(s_{t+1}, a_{t+1}^1, \cdots, a_{t+1}^N)\big|_{a_{t+1}^i \sim \pi_{\theta'^i}(\cdot\,|\,s_{t+1})},$$

12:           Update critic by minimizing the loss $\mathcal{L}(\omega^i) = \frac{1}{S}\sum_{t=1}^S \left(y_t - \bar{Q}_{\omega^i}(s_t, a_t)\right)^2$
13:           Update actor using the sampled policy gradient

$$\nabla_{\theta^i} J^i(\theta) \approx \frac{1}{S}\sum_{t=1}^S \nabla\pi_{\theta^i}(a_t^i\,|\,s_t)\nabla_{a^i}\bar{Q}_{\omega^i}(s_t, a_t^1, \cdots, a^i, \cdots, a_t^N)\big|_{a^i=\pi_{\theta^i}(s_t)},$$

$$\theta'^i = (1-\tau)\theta^i + \tau\nabla_{\theta^i} J^i(\theta)$$

$$\nabla_{\theta^{0,i}} J^i(\theta) \approx \frac{1}{S}\sum_{t=1}^S \nabla\pi_{\theta^{0,i}}(s_t)[a_t] + \eta\sum_{t=1}^S \nabla\left(\pi_{\theta^{0,i}}(s_t)[a_t] - \bar{r}_t^i\right)^2,$$

$$\theta'^{0,i} = (1-\tau)\theta^{0,i} + \tau\nabla_{\theta^{0,i}} J^i(\theta)$$

14:         **end for**
15:     **end for**
16: **end for**

---

where $\mu \in [0, 1]$ denotes the uncertainty level, and $P_{\theta^{0,0}}^{\texttt{pert}}(\cdot\,|\,s, a)$ is some perturbation of the transition, which is parameterized by $\theta^{0,0}$. In this case, (B.8) becomes

$$\nabla_{\theta^{0,0}} \bar{V}_{\tilde{\pi}_\theta}^i(s)$$
$$= \sum_{s'\in\mathcal{S}} \rho_{\pi_\theta}^s(s') \cdot \mathbb{E}_{a'\sim\pi_\theta(\cdot\,|\,s')}\left[\sum_{s''\in\mathcal{S}} \gamma\mu \cdot \nabla P_{\theta^{0,0}}^{\texttt{pert}}(s''\,|\,s', a') \cdot \bar{V}_{\tilde{\pi}_\theta}^i(s'')\right]$$
$$= \sum_{s'\in\mathcal{S}} \rho_{\pi_\theta}^s(s') \cdot \underbrace{\mathbb{E}_{a'\sim\pi_\theta(\cdot\,|\,s'), s''\sim P_{\theta^{0,0}}(\cdot\,|\,s', a')}\left[\gamma\mu \cdot \nabla\log P_{\theta^{0,0}}^{\texttt{pert}}(s''\,|\,s', a') \cdot \bar{V}_{\tilde{\pi}_\theta}^i(s'')\right]}_{\texttt{can be sampled without knowing the model}}, \quad \text{(B.9)}$$

which might be easier to implement in simulations.

*Proof of Lemma 3.1:*

The results can be obtained from Lemma B.1 by simply fixing $P_{\theta^{0,0}}$ as $P$, and re-defining the corresponding Q-value and state-value functions following (3.5) and (3.4) instead. $\qquad\square$

## C   Actor-Critic for Robust MARL

As also stated in [13], if working with deterministic policies, one can write the first gradient in Lemma 3.1 as:

$$\nabla_{\theta^i} J^i(\theta) = \mathbb{E}_{s,a\sim\mathcal{D}}\left[\nabla\pi_{\theta^i}(a^i\,|\,s)\nabla_{a^i}\bar{Q}_{\tilde{\pi}_\theta}^i(s, a)\big|_{a^i=\pi_{\theta^i}(s)}\right], \quad\quad\quad \text{(C.1)}$$

where $\mathcal{D}$ is the replay buffer containing the samples $(s, a^1, \cdots, a^N, s', \bar{r}_s^1, \cdots, \bar{r}_s^N)$ collected from experiences of all agents. We consider the deterministic policies in our experiments, and use (C.1)

Figure 4: The particle environments used in our experiments. From left to right: "cooperative navigation", "keep-away", "physical deception", and "predator-prey". The figure is based on [13].

when developing our Robust-MADDPG method, as stated in Algorithm 1. Note that the minimization in line 12 in Algorithm 1 can be solved by any non-convex optimization solvers, or by just several iterations of gradient steps w.r.t. $\omega^i$. In addition, in the second update (nature policy update) of Line 13, we add another term to restrict the nature policy output in the uncertainty set.

## D    Environments

We use the similar particle environments as in [13], where there is a two-dimensional world with continuous space and discrete time. The agents can only perform physical actions. More specifically, we consider the following four environments. One can also refer to Figure 4 for graphical illustrations of the experimental scenarios.

**Cooperative navigation.**    In this cooperative environment, $N$ agents collaborate to occupy $N$ landmarks. Each agent observes the relative positions of other agents and landmarks. Moreover, agents are rewarded based on the minimum distance of any agent from each landmark, and are penalized if they collide with other agents.

**Keep-away.**    The fully competitive environment consists of $L$ landmarks including a target land-mark, an agent who knows the target, and an adversarial agent whose goal is to push the agent away from the target. The loss of the agent is the distance to the target landmark, and the adversary is rewarded by occupying the goal while keeping the agent away, although the adversary does not know the correct target. This must be inferred from the agent's movements.

**Physical deception.**    This mixed cooperative-competitive environment has $N$ landmarks with a target landmark. An adversarial agent has no information about the target landmark, but attempts to find out and occupy it. The reward of adversary is simply based on how close it is to the target. In addition, another $N$ agents are trying to 'deceive' the adversary by cooperatively reaching the target while also occupying other landmarks. The agents are rewarded by how close the nearest one is from the target landmark, and also penalized by the distance of the adversary to the target.

**Predator-prey.**    In this scenario, $N$ slower cooperating predators aim to hit a faster prey agent around the world with $L$ randomly generated obstacle landmarks that block the way. Predators get rewards every time one of them touches the prey, and the prey gets penalized in the mean time. Each agent observes the relative positions and velocities of other agents and positions of the obstacles.

## E    Implementation Detail

We implemented Robust-MADDPG in PyTorch based on the source code of Recurrent MADDPG[5]. We also re-implemented M3DDPG based on its authors' source code[6]. All actors, critics, and the nature actors use the same two-layer MLP architecture. Each agent either knows other agents' actors or needs to approximate them during learning. In our experiments, we reported the results assuming each agent knows all the other actors. Since some environments are intrinsically more difficult than

others, we adopt different numbers of training episodes to allow for convergence. For keep-away and physical deception, we trained for 50K episodes. For cooperative navigation and predator-prey, we trained for 100K episodes. All experiments were run on AWS EC2 p3.x16 instances where each 100K training episodes job took about 5-5.5 hours. We used the default hyperparameters in the source code. One new hyperparameter we added is the MSE weight $\eta$ in Line 13 of the algorithm. When $\eta$ is low (such as 0.001 or 0.01), the MSE part weights too little and cannot constrain the nature policy to fall into the uncertainty set. After tuning, we found $\eta = 0.1 \sim 1$ is generally a good range but we want to emphasize it is task-dependent.

## Footnotes

[5] https://github.com/nicoring/rec-maddpg

[6] https://github.com/dadadidodi/m3ddpg