[Reviews · NeurIPS 2020]

Review 1

Summary and Contributions: This paper proposes a new robust Multi-agent RL based framework, which models reward function and transition probability to achieve robust learning in the environment. Besides, they propose a new formalization of Nash Equilibrium and give it a theoretical guarantee of convergence.

Strengths: The paper introduces a new MARL based framework which modelling the uncertainty of environments by setting a ‘nature’ agent, which modelling individual reward and transition function of each agent. The ‘nature’ agent aims at considering the worst case in an uncertain environment by minimizing environment functions, which help formalize uncertain general-sum Markov game. The theoretical analysis of convergence improves the soundness of this method. Experiments: For a paper that introduces a new Nash Equilibrium, empirical performance is particularly important as it seems otherwise hard to optimize. It uses a variant MADDPG method to optimize the problem, which reduces the complexity of optimization.

Weaknesses: 1. As the paper claimed, the uncertainty is modelled based on individual reward function and transition probability. It seems ref [12] has the same definition of reward and transition function (eq.3 in ref[12]), but they do not use the description like ‘nature’ agent and they use minmax to formalize the objective. This paper looks like another application of [12] and use a different optimizing trick. As such, it is critical to provide detailed discussion about the difference between the proposed method and ref [12]. 2. Introducing MADDPG (section 3.2) to optimize the hard objective is certainly of interest. The paper gives the gradient of policy. However, it might be better to give the theoretical guarantee of the discrepancy between the learned policy and true policy as well as the equilibrium guarantee by using such an optimization framework (MADDPG). 3. It might be better to report standard errors in tables. Experiment part lacks details (like the number of agents etc.) and experimental environments are too simple to support the effectiveness of your method. It is better to extend the experiment to more complex environments with multiple kinds of uncertainty.

Correctness: The method seems correct and the derivations are rigorous.

Clarity: The paper is well organized and written.

Relation to Prior Work: The description is almost clear, but it is better to illustrate the different between ref [12]. Otherwise, the novelty is questionable.

Reproducibility: Yes

Additional Feedback:


Review 2

Summary and Contributions: Based on rebuttal and discussion: Upon reading R1 and the author rebuttal, as well as carefully referring to the article again, I recognise that the definition of robust Markov games has been taken literally from the literature [12], and the novelty is thus rather in employing this framework in the context of RL. This reduces the novelty of the contribution. I agree with the shortcomings mentioned in other reviews that - the re-use of “Robust Stochastic Games” could have been more emphasized (albeit they do cite [12] in appropriate places); and differences from [12] could have been expounded more. - the empirical evaluation has a character that is rather illustrative than complete. While I agree with R4 that the motivating example for robustness is not perfect, I see the authors’ attempt to juxtapose (“not only… but …” in L107f) the two challenges of responding to unknown opponent policies (general in MARL) and responding to uncertainty in models (specific to robust MARL). In my view it remains a very solid account of robust MARL, making a timely contribution to an important topic. It is the kind of paper that transfers a new angle of robustness that is solidly established in control theory with a lot of clarity to the MARL community. --- This article transfers the concept of robustness to Markov games, formalizing the new concept of robust Markov games. This decision-making framework captures distribution-free model discrepancies in multi-agent interactions. The authors establish existence of equilibria, and provide a Q-learning algorithm with convergence guarantees and an actor-critic algorithm (extending MADDPG towards robustness) with empirical comparisons to state of the art.

Strengths: The contributions are well developed and timely, as robustness has received increasing attention in the multi-agent context. The article is comprehensive, and expounds on all aspects from the new extended formal task (Robust Markov Game), via algorithms to their theoretical convergence and empirical exposition of performance.

Weaknesses: If one was looking for opportunities of improvement, minor clarifications could be added (see clarity and additional comments), and the interpretation of natures moves as optimal attacks (if the underlying task could be disrupted externally) could be discussed. The empirical evaluation is ok, but not as outstanding as the theoretical part of the paper. In particular, results are very dense and take time to parse, and the use of a distributional setting is not motivated (despite the distribution-free appraoch of robustness). A comparison to a Bayesian omniscient upper bound solution could have added more context to the amount of performance sacrificed to not knowing the distribution. Yet, in the context of space restrictions I understand the authors restriction to their chosen setting.

Correctness: The article proceeds to present arguments and derivations meticulously, and claims are sound.

Clarity: Despite covering a lot of ground, arguments are presented concisely and precisely, making it a breeze to follow this article. A notable exception is the introduction of $\theta^0$ or "the set of natures policies", where a few words could be added to make explicit that nature moves against each agent.

Relation to Prior Work: The article cites appropriate sources, and provides a sharp demarcation to important related work.

Reproducibility: Yes

Additional Feedback: - In contributions, the last sentence mentions "model uncertainties" but could re-iterate "distribution-free"; since this is a pivotal point to avoid confusion with distributional uncertainties (allowing Bayesian approaches and being covered by related work). - "The results, though not generally apply..." grammar - "which is indeed a variant of [...] (MADDPG) method" either add "the" or remove "method" - Within Section 4, Keep-away: Is "significantly better" verified statistically? - Within Physical deception, the paragraph could provide the annotation for Table 2 evaluation metric abbreviations; those seem currently undefined (especially AG/ADV) even though explanations seem to correspond.


Review 3

Summary and Contributions: This paper studies the multi-agent reinforcement learning under model uncertainty, which involves additional partial 'noise' / 'randomness' in agents' reward functions and transition function. Based on that setting, the authors define robust Markov perfect Nash equilibrium. Under the different assumptions, the authors investigated the tubular value iteration algorithm, which knows the environment model. Moreover, a practical parameterized AC algorithm to deal with massive or even continuous state-action spaces problems, where 'nature policies' play against the agents, which learn/approximate the agents' reward given the state from experience.

Strengths: The authors have investigated an interesting robust Markov game problems and provided both theoretical justifications and a practical algorithm.

Weaknesses: 1. The formulation of the robust Markov game needs more details, as i mentioned above. 2. I found the the 'nature policy' plays an important role in the algorithm, but the authors only have limited word about how to approximate a good 'nature policy'. For example, the eq 3.5 only minimizes the reward, I can confused what is the learning signal to recover the original reward. And in Algo 1, line 13, an additional MSE loss is introduced for 'nature policies' objectives without the definition of eta, I think this would be important for the success the algorithm, authors need to make more clarification and evaluation in the main paper.

Correctness: The results should be ok but need to be further evaluated.

Clarity: The paper is easy to follow, but some typos need to be corrected. And, I would suggest more descriptions of the difference between Robust Markov Games and Markov Games with stochastic reward/transition. Besides, a mathematical description of the uncertainty is welcome, like in [1]. [1] Lim, Shiau Hong, Huan Xu, and Shie Mannor. "Reinforcement learning in robust markov decision processes." Advances in Neural Information Processing Systems. 2013.

Relation to Prior Work: The authors provided sufficient discussion about the connections between previous MARL work and robust RL/MDP. But some works about dealing with non-stationary MDP using model-based methods are missing, which are highly related to this work.

Reproducibility: Yes

Additional Feedback: The formulation/algorithm related concerns are listed in the above. The experiments is relatively weak. The authors have done the experiments on several particle environments using MADDPG and M3DDPG as baseline. I noticed that the DDOG is also implemented in the code, why the authors don't include it as baselines. Besides, the uncertainty would bring more variance, it would be good if authors can add the std in the main text's experiment results. I see the learning curves in Fig 5, but what larger noise=7.0 has lower variance during the learning comparing to the noise=6.0/5.0? Finally, as I mentioned, the nature policy is important in your paper, it would be good to set more ablation study on how nature policy works. For example, adding a setting: during the training the reward always to be the minimum in the valid range, and test in original uncertainty.


Review 4

Summary and Contributions: - The paper extends the robust MDP framework to robust Markov games in the multi-agent case. - The paper proposes Q-learning and actor-critic algorithms for the robust MARL case - The paper evaluates their robust MADDPG (R-MADDPG) algorithm on some particle world environments with varying degrees of reward noise.

Strengths: - The robust Markov game framework seems like a potentially useful concept. - The paper proposes both Q-learning and actor-critic algorithms, including a practical algorithm in the function approximation case (R-MADDPG). - In experiments with moderate to high reward noise, the R-MADDPG algorithm seems to outperform the baselines that don’t factor in reward noise. - I appreciate that the paper compares to M3DDPG, another robust MARL method.

Weaknesses: - The biggest weakness of this paper in my mind is the clarity and framing. The paper motivates the contribution by stating that agents may not have access to the reward functions / models of other agents. For example, the paper states: “In many practical applications, the agents may not have perfect information of the model, i.e., the reward function and/or the transition probability model. For example, in an urban traffic network that involves multiple self-driving cars, each vehicle makes an individual action and has no access to other cars’ rewards and models.” However, most MARL methods don’t make any assumptions about the reward function of other agents, particularly in the decentralized MARL setting. So it should be clarified that the robust MARL framework additionally handles uncertainty in the agent’s own reward function (which is potentially useful, and which decentralized MARL does not do). This and other clarity issues significantly impact a readers’ understanding of the paper. - The proposed R-MADDPG algorithm performs notably worse than the baselines in the case where there is no reward noise. The paper states: “We observe that when the uncertainty level is zero or low, non-R-MADDPG models perform better overall, because the nature policy in R-MADDPG brings model errors in selecting rewards.“ I’d like this to be explained in more detail. Specifically, when entering a new environment it’s unknown how much noise there will be in the observed reward function --- it’s unclear whether the reward noise in realistic environments would be sufficient for R-MADDPG to be useful. - The results only show the average of 5 runs, with no error bars. I imagine these results may have quite a bit of variance, so this is a bit concerning. - “Without loss of generality, we follow the convention as in [12], and only focus on the uncertainty in the reward function hereafter for simplicity.” This seems like a fairly important step, since the paper was motivated by considering model uncertainties, so I think this should be explained in a lot more detail.

Correctness: Other than the framing issue mentioned above, the methodology appears correct to me, though I did not look in detail into proofs, etc.

Clarity: This is a weakness of the paper, as outlined above.

Relation to Prior Work: This is done fairly well.

Reproducibility: Yes

Additional Feedback: With the concerns about framing, clarity, and results stated above, I believe this paper is borderline, with a tendency towards rejection. Other notes: -It’s a bit strange that a proof of proposition 2.2 isn’t really provided (even though it’s straightforward). Grammatical errors: - L44: affects -> effects - L117: a prior -> a priori ------- Update after author feedback: I appreciate the addition of the confidence intervals, and the clarification of the framing. I am updating my review to a 6 given these additions. I still have some concerns about the general framing of the paper (e.g. R1's point about the robust Markov game framework coming from [12]) and, while the authors say they will address this in an updated version, it is difficult to determine how well this will be done without actually seeing an updated version.

[Author Response · NeurIPS 2020]

**Rev. 1** **1)** Yes, the formulation of "robust Markov game" originated from [12]. But our contribution is to model the "robust MARL" problem as such a game and develop algorithms to solve it, both of which are original. Introducing the "nature" agent is exactly one contribution that makes the computation tractable and easier to understand in MARL context. It is not simply a "trick", but our effort to transfer [12] to robust MARL. We will add more clarification on this. **2)** With general function approximation, PG is solving a non-convex optimization problem, the theoretical guarantee of its optimality is very challenging. In fact, the optimality guarantee of PG methods, even in the single-agent setting, can only be established in limited cases until recently. Note that the equilibrium guarantee of the original MADDPG is non-existent yet. As our work is the first one that dealt with robust MARL under the framework of robust Markov games, many other aspects, e.g., the algorithm design, solution existence, Q-learning convergence, and empirical validation, need to be done first . **3)** We had put experiment details in the supplement due to page limit. We will re-organize the experiment session to add those details and include results for the cooperative navigation in the main content. We have updated all the tables with 95% confidence intervals across 5 trials, 1000 evaluation episodes for each. Below is an example for keep-away. The figures for the cooperative navigation in the supplement have also shown 95% CI across five runs. We hope our response can help address your concerns on the novelty, and re-evaluate our submission.

**Rev. 2** Thanks for the very positive feedback, we will address the grammatical typos and the notations in Table 2, and add more clarification on the nature agent. **1)** We will re-iterate the "distribution-free" point and add clarification to avoid the possible confusion with "distributional uncertainties". **2)** For Keep-away, we have run student's t tests. All calculated $p$-values are below 0.01, suggesting that the improvements are statistically significant.

**Rev. 3** **1)** We are not sure what was meant by "as I mentioned above", as there was only the reviewer's summary above it . **2)** The "nature" policy is approximated by neural networks, as for other agents. We are confused about the comment "eq 3.5 only minimizes the reward", since Eq. 3.5 has no minimization. There is no "original reward" in our robust MARL setting, as the reward function is "uncertain". Instead, the nature agent is always "playing against" all agents, while the agents want to find the policy that account for this adversarial uncertainty. The MSE in line 13 is some heuristic to regularize the nature policy's output, with $\eta$ being the weight coefficient. We have mentioned the choice of $\eta$ in Sec. F (appendix),

| Models | | Uncertainty level ($\lambda$) | | | |
|---|---|---|---|---|---|
| AG | ADV | 0.0 | 1.0 | 2.0 | 3.0 |
| M3 | M3 | **13.23 (0.12)** | 6.34 (0.18) | 3.01 (0.14) | 1.34 (0.09) |
| M3 | MA | 8.07 (0.18) | **7.51 (0.17)** | 3.47 (0.14) | 2.86 (0.13) |
| M3 | RM | 13.15 (0.12) | 7.23 (0.18) | **5.63 (0.17)** | **4.31 (0.16)** |
| MA | M3 | **13.57 (0.11)** | **8.12 (0.18)** | 2.58 (0.13) | 1.66 (0.09) |
| MA | MA | 10.91 (0.16) | 6.04 (0.17) | 3.89 (0.15) | 1.44 (0.09) |
| MA | RM | 12.52 (0.13) | 6.96 (0.17) | **4.05 (0.15)** | **3.47 (0.14)** |
| RM | M3 | **11.87 (0.15)** | 6.98 (0.18) | **5.65 (0.16)** | 2.09 (0.11) |
| RM | MA | 8.83 (0.17) | 7.38 (0.18) | 4.37 (0.16) | 3.25 (0.14) |
| RM | RM | 7.89 (0.18) | **8.21 (0.18)** | 4.32 (0.16) | **3.61 (0.14)** |

Table 1 in the paper updated with 95% confidence interval.

and will add more clarification in the main paper. **3)** Clarity: We will address the typos. The "robust Markov games" is fundamentally different from "Markov games with stochastic reward/transition". The latter is nothing but the standard MG model, since it allows the reward/transition to be stochastic. However, our robust MG model accounts for the "model uncertainty" explicitly, like the single-agent setting in [1]. The mathematical description of uncertainty is explicitly provided in the robust MG formulation. **4)** Related work: We note that they are not that relevant, as non-stationary MDP usually deals with an online/adaptive setting, with the focus of reducing the accumulated regret, while we account for the "model uncertainty" from the beginning, and our algorithms are model-free . Most importantly, our main focus is on the "multi-agent" side. **5)** The MADDPG paper [19] has already shown that MADDPG outperforms DDPG in all these particle environments. There is no added value to compare to DDPG again. **6)** About the variance, please check our response for reviewer 1, answer 3. Since we run evaluation over 1000 episodes for each trial, the variance, i.e., CI, is fairly acceptable. **7)** We have the same intuition that the confidence interval for high noise level (7.0) should be larger than 5.0/6.0, we think we get the narrower CI only by chance. **8)** We have started running the suggested ablation experiments that the nature always selects the minimum reward in an uncertainty set. But the training takes much longer for convergence and we could not obtain the final results before deadline. We will add it to the final version. We notice that most of your comments are regarding the clarification, and the "weakness" part seems not that fatal. We sincerely hope our response can help address your concerns, and re-evaluate our paper.

**Rev. 4** **1)** Clarity and framing: our focus was "the agents may not have perfect information of the model" from each agent's perspective, including both *its own* and *others'* model. Sorry for the confusion in the traffic network example, we will clarify on this point. **2)** When there is no reward noise, non-R-MADDPG methods can obtain the exact reward, while R-MADDPG still uses the nature agent to approximate the certain reward, which leads to extra approximation errors. However, R-MADDPG does not always "perform notably worse" than the baselines. E.g., in the keep-away experiments ($\lambda = 0$ or 1), R-MADDPG adversary performs better than MADDPG most of the time, and slightly worse than M3DDPG. In practice, when entering a new environment, we suggest running both algorithms and compare. **3)** Please check our response to reviewer 1, answer 3 and reviewer 3, answer 6 on the variance concern. **4)** We will explain the "Without loss of generality..." sentence with more details in revision. **5)** We will provide the proof of Proposition 2.2 for completeness. As the main weakness was on clarify and framing, which can be addressed as per your suggestion easily, we really appreciate the reviewer to re-evaluate our contribution, especially considering that our work provides the first framework that accounts for "model-uncertainty" in MARL, with both theory and simulations.

[Meta-Review · NeurIPS 2020]

The authors' feedback resolved some of the concerns raised by the reviewers. Unfortunately, we have not been able to reach a consensus, so this paper is borderline. On the positive side, the paper introduces a new and interesting MARL framework and provides both theoretical and practical contributions. On the negative side, the authors should better highlight from the beginning what is already present in the state of the art and where their contributions start from, and they should provide a more extensive and accurate empirical evaluation. The requested changes are quite significant, but given the authors' rebuttal, I feel they can fix these issues and so I suggest acceptance.